# Shaping the physical world to our ends through the left PF technical-cognition area

François Osiurak[1,2]*, Giovanni Federico[3], Arnaud Fournel[1], Vivien Gaujoux[1], Franck Lamberton[4], Danièle Ibarrola[4], Yves Rossetti[5,6], Mathieu Lesourd[7]*

[1]Laboratoire d'Étude des Mécanismes Cognitifs, Université de Lyon, Bron, France; [2]Institut Universitaire de France, Paris, France; [3]Laboratory of Experimental Psychology and Cognitive Neuroscience, Suor Orsola Benincasa University, Naples, Italy; [4]CERMEP-Imagerie du vivant, MRI Department and CNRS UMS3453, Lyon, France; [5]Centre de Recherche en Neurosciences de Lyon (CRNL), Trajectoires Team (Inserm UMR_S 1028-CNRS-UMR 5292-Université de Lyon), Bron, France; [6]Mouvement et Handicap and Neuro-Immersion, Hospices Civils de Lyon et Centre de Recherche en Neurosciences de Lyon, Hôpital Henry Gabrielle, St Genis Laval, France; [7]Université Marie et Louis Pasteur, INSERM, UMR 1322 LINC, Besançon, France

**\*For correspondence:**
francois.osiurak@univ-lyon2.fr (FO);
mathieu.lesourd@univ-fcomte.fr (ML)

**Competing interest:** The authors declare that no competing interests exist.

## eLife Assessment

This **valuable** study used functional MRI experiments to identify the involvement of a left parietal area (PF) in reasoning about the physical properties of actions, objects, and events. **Solid** evidence was shown regarding the commonalities and differences across different types of reasoning tasks, yet the methodological and theoretical interpretations require further scrutiny. The study would be of interest to researchers studying the cognitive and neural mechanisms of reasoning and problem solving.

**Abstract** Our propensity to materiality, which consists in using, making, creating, and passing on technologies, has enabled us to shape the physical world according to our ends. To explain this proclivity, scientists have calibrated their lens to either low-level skills such as motor cognition or high-level skills such as language or social cognition. Yet, little has been said about the intermediate-level cognitive processes that are directly involved in mastering this materiality, that is, technical cognition. We aim to focus on this intermediate level for providing new insights into the neurocognitive bases of human materiality. Here, we show that a technical-reasoning process might be specifically at work in physical problem-solving situations. We found via two distinct neuroimaging studies that the area PF (parietal F) within the left parietal lobe is central for this reasoning process in both tool-use and non-tool-use physical problem-solving and can work along with social-cognitive skills to resolve day-to-day interactions that combine social and physical constraints. Our results demonstrate the existence of a specific cognitive module in the human brain dedicated to materiality, which might be the supporting pillar allowing the accumulation of technical knowledge over generations. Intensifying research on technical cognition could nurture a comprehensive framework that has been missing in fields interested in how early and modern humans have been interacting with the physical world through technology, and how this interaction has shaped our history and culture.

## Introduction

Modern societies heavily rely on and even depend upon technology. From the first lithic industries to the most modern and complex ones, technologies enable us to fulfil elementary needs such as feeding or building shelters, and even more sophisticated ones such as communicating, conquering space, or playing music. Human technology is ubiquitous and this omnipresence reflects how we, as a species, are remarkably skilled at shaping the physical world according to our needs (*Boyd et al., 2011*; *Lombard, 2016*). A paradox though remains in modern science. Even if this special skill has allowed our prolific expansion, only a little attention has been paid to the cognition needed to practically develop the technology required (*Whiten, 2011*; *Wynn et al., 2017*; *Wynn and Coolidge, 2014*; *Whiten, 2022*; *van Elk, 2021*). We do make and use advanced technologies, and we also transmit them to the next generations, but the full understanding of the neurocognitive processes implied is still in its infancy.

Most of this understanding has long come from clinical neuropsychology, through the observation of apraxic patients in whom damage to the left inferior part of the parietal lobe leads to tool-use disorders (*De Renzi and Lucchelli, 1988*; *Goldenberg and Hagmann, 1998*). These disorders have been initially interpreted as reflecting impaired sensorimotor programs dictating the prototypical manipulation of common tools (e.g., a hammer) (*van Elk et al., 2014*; *Heilman et al., 1982*). This manipulation-based approach has provided interesting insights (*Kleineberg et al., 2022*; *Weiss et al., 2016*; *Kroliczak et al., 2021*; *Michalowski et al., 2022*; *Wheaton et al., 2009*) and even elegant attempts to explain how these sensorimotor programs could support the use of both unfamiliar and novel tools (*Buchmann and Randerath, 2017*; *Mizelle and Wheaton, 2010*; *Stoll et al., 2022*; *Seidel et al., 2023*), but remains silent on the more general cognitive mechanisms behind human technology that include the use of common and unfamiliar or novel tools but must also encompass tool making, construction behaviour, technical innovations, and transmission of technical content.

This silence has been initially broken by a series of studies initiated by *Goldenberg and Hagmann, 1998*, which has documented a behavioural link in left brain-damaged patients between common tool use and the ability to solve mechanical problems by using and even sometimes making novel tools (e.g., extracting a target out from a box by bending a wire to create a hook) (*Goldenberg and Hagmann, 1998*; *Heilman et al., 1997*). Brain-lesion studies have revealed that this behavioural link has a neural reality because both common and novel tool uses are impaired after damage to the left inferior parietal lobe and particularly the area PF (*Goldenberg and Spatt, 2009*; *Martin et al., 2016*). As (*Goldenberg and Spatt, 2009*) claimed, '[t]hese results support the conclusions that the parietal lobe contribution to tool use concerns general principles of tool use rather than knowledge of the prototypical use of common tools and objects, and the comprehension of mechanical interactions of the tool with other tools, recipients or materials rather than the selection of grip formation and manual movements' (p. 1653). Neuroimaging studies have thereafter extended *Goldenberg and Spatt, 2009* conclusion to situations other than tool use strictly speaking. For instance, evidence has indicated the preferential activation of the left area PF when people observe others use tools (*Reynaud et al., 2019*) as well as when people view physical events, whether they are instructed or not to reason about them (*Fischer et al., 2016*; *Pramod et al., 2022*). Also, the cortical thickness of the left area PF was found to predict the performance on psychotechnical tests (*Federico et al., 2022*) (e.g., water-pouring problems). It is noteworthy that these studies have also reported the involvement of other areas, such as the left inferior frontal gyrus (IFG) or the lateral occipitotemporal cortex (LOTC).

These findings have fuelled the development of the technical-reasoning hypothesis (*Osiurak and Reynaud, 2020*; *Osiurak et al., 2023*), which offers a larger account of the neurocognitive processes at work for understanding and shaping our physical world and for successfully passing technologies to the next generations in a cumulative manner, forming what has been dubbed the cumulative technological culture (*Boyd et al., 2011*; *Tomasello et al., 1993*). Technical reasoning refers to the ability of reasoning about the physical properties of objects and is nurtured by implicit mechanical knowledge acquired through interactions with objects. It is both causal (i.e., prediction of future events) and analogical (i.e., transfer from one situation to another). To solve a physical problem, an agent can use prior knowledge about mechanical properties and physical laws that apply to the physical world, such as gravity and leverage. Then, this knowledge is combined with the constraints of the current situation, which include the available data, and any objects required to achieve the desired goal. The outcome of the technical-reasoning process is a simulation of the mechanical action to be

performed, which then constrains, if the individual intends to act, their bodily actions through a kind of mechanical-to-motor action cascade, which will ultimately feedback onto the central representations of our body, space, and objects. As mentioned above, the left area PF occupies a central place in the technical-reasoning network, along with additional brain areas such as the left IFG and LOTC.

Here, we focus on two key aspects of the technical-reasoning hypothesis that remain to be addressed: generalizability and specificity. If technical reasoning is a specific form of reasoning oriented towards the physical world, then it should be implicated in *all* (the generalizability question) *and only* (the specificity question) the situations in which we need to think about the physical properties of our world. To tackle these two questions, we designed two fMRI experiments that included four different tasks (Experiment 1: Mechanical problem-solving task; *n* = 34; Experiment 2: Psychotechnical task, fluid-cognition task, and mentalizing task; *n* = 35; *Figure 1* and *Figure 1—figure supplement 1*), detailed in the following lines.

The first question concerns the generalizability of the technical-reasoning network to any context that includes physical understanding. As suggested above, neuropsychological studies have supported *Goldenberg and Spatt, 2009* conclusion about the contribution of the inferior parietal lobe to general principles of tool use, but no neuroimaging study has confirmed this conclusion so far. For this, we designed a first experiment allowing the observation of the cerebral activities related to physical problem-solving implying tool use. Participants were presented with mechanical problems, consisting in figuring out how to move, with the help of a novel tool, a small red cubic element trapped in a 3D glass box projected on a screen from its original location into a new target location (*Figure 1A*). This task allowed us to test the involvement of the technical-reasoning network – and particularly of the left area PF – in novel tool use. Nonetheless, the generalizability of technical reasoning implies that its network is recruited beyond tool use and serves as a basis for the physical understanding of the world surrounding us. We tested this assumption by studying the neural correlates tied to the understanding of physical principles, disembodied from any tool-use situation. In a second neuroimaging experiment, participants performed a psychotechnical task in which they had to solve non-tool-use physical problems, such as water-pouring problems (*Hegarty, 2004*; *Figure 1B*). We predicted that the technical-reasoning network and particularly the left area PF should be recruited to perform this psychotechnical task. In addition, as technical reasoning is supposedly a central component of the mechanical problem-solving task and the psychotechnical task, and the INT + PHYS and PHYS-Only conditions of the mentalizing task, which will be described below, we hypothesized that the technical-reasoning network should be commonly activated across these tasks and should be found in a conjunction analysis of the four experimental conditions.

The second question concerns the specificity of the technical-reasoning network. As technical reasoning is an implicit and causal form of reasoning, it may be easily conflated with other forms of implicit/non-verbal and/or causal reasoning, such as fluid reasoning or mentalizing. Thus, in the second experiment, we tested its specificity by asking participants to perform fluid-reasoning and mentalizing tasks. Fluid reasoning, hereafter called fluid cognition, refers to temporarily maintaining information to produce adapted responses to solve novel problems or plan and execute directed behaviour based on inductive and deductive relationship (*Blair, 2006*). Although fluid cognition is a non-verbal form of reasoning, it does not necessitate knowledge of the physical world, its constraints and the mechanical laws governing it. The distinction between technical reasoning and fluid cognition has already been supported by behavioural evidence (*De Oliveira et al., 2019*) as well as by neuroimaging studies that have shown the recruitment of the prefrontal cortex in fluid cognition (*Hobeika et al., 2016*), with brain regions that are not commonly reported in the context of tool use or physical understanding (e.g., dorsal prefrontal cortex and medial superior frontal gyrus). The participants in the second experiment had to complete a fluid-cognition task, which was an adaptation of the Raven's Progressive Matrices test (*Figure 1C*). This test has been widely used to predict performance on a wide range of logical reasoning tasks and has been found to predict the general factor of intelligence (*Gray and Thompson, 2004*). We hypothesized the recruitment of the fluid-cognition network, particularly the prefrontal cortex, in the fluid-cognition task, which should diverge from the network involved in the psychotechnical task. Mentalizing refers to detecting and attributing mental states to others or oneself (*Gallagher and Frith, 2003*; *Van Overwalle and Baetens, 2009*). It is a form of causal reasoning, given that it can be used to infer how hidden mental states can cause some specific behaviours. Depending on the situation, this ability requires the collaboration of several cognitive

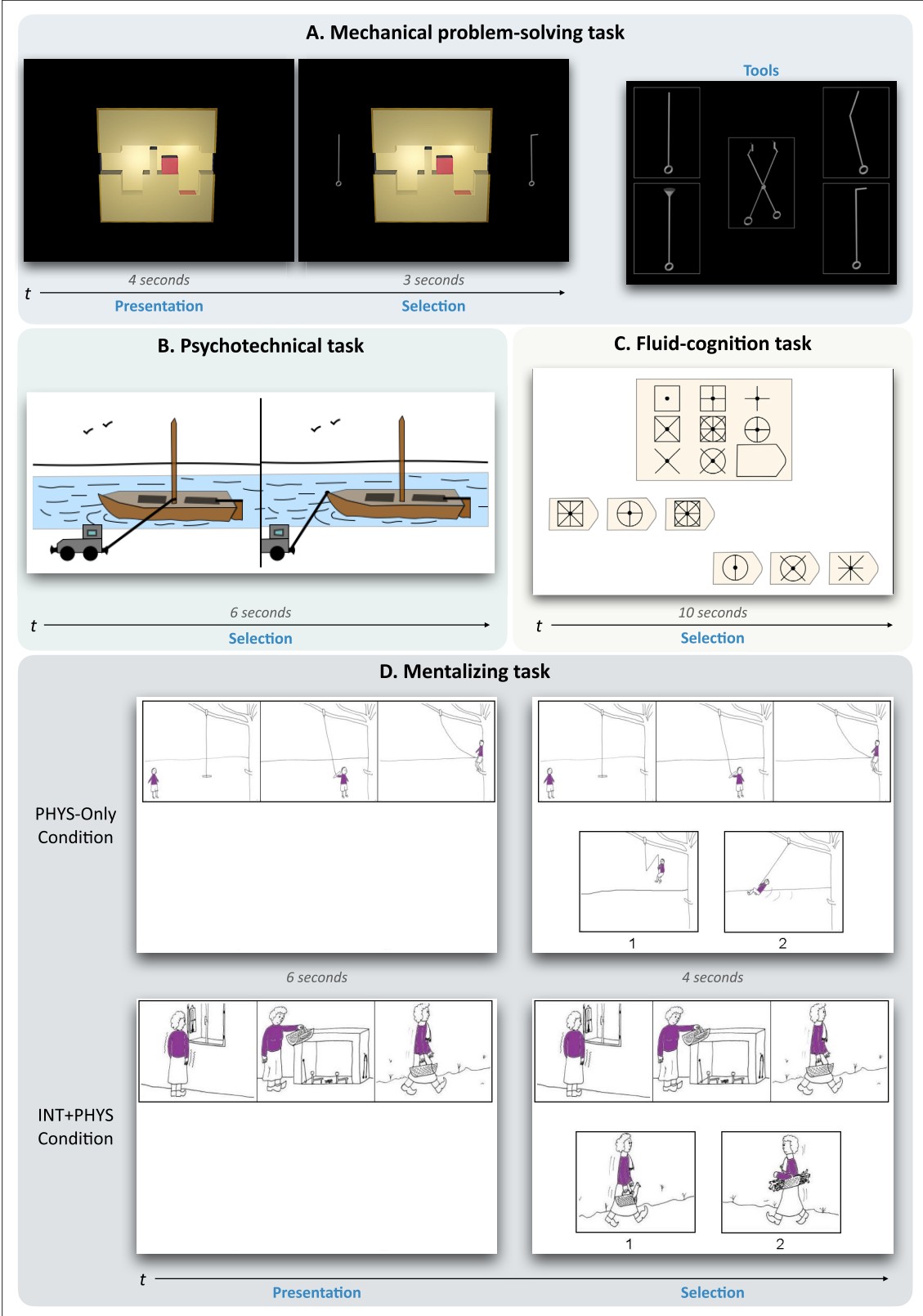

**Figure 1.** Experimental tasks. (**A**) In the mechanical problem-solving task (Experiment 1), participants had to figure out for 4 s how to move a red cube trapped in a 3D glass box from its original location to a new target location. Then, two tools were presented for 3 s, and they had to decide which was the correct one to solve the mechanical problem. Before the scanning session, the participants were informed that five distinct tools could be used to solve the mechanical problems. (**B**) In the psychotechnical task (Experiment 2), two situations were displayed for 6 s. Participants had to select which of

*Figure 1 continued on next page*

*Figure 1 continued*

the two displayed situations was the correct one or the most effective one. (**C**) In the fluid-cognition task (Experiment 2), the participants had to select the line of options with the correct one. (**D**) In the mentalizing task (Experiment 2), the superior part of the board was shown for 6 s, for the participants to try to make sense of the cartoon first. Then the bottom part was presented for 4 additional seconds, with the top part remaining on display. The participants had to choose the cartoon with the probable ending to the story depicted in the three first drawings. In the PHYS-Only condition, the selection only needed to understand the physical context. In the INT + PHYS condition, the selection needed to understand both the physical context and the social context. Illustrations in (**D**) reproduced with permission from Birgit Völlm. No permission was needed for the pictures in (**A**) as we built this task. The items of the psychotechnical task (**B**) and the fluid-cognition task (**C**) are adapted from commercialized tests and do not correspond to the original items of these tests. For more information, see the Methods section.

The online version of this article includes the following figure supplement(s) for figure 1:

**Figure supplement 1.** Control conditions of the experimental tasks.

mechanisms (*Gallagher and Frith, 2003*; *Van Overwalle and Baetens, 2009*), such as perspective taking (medial prefrontal cortex), the understanding of communicative gestures (temporoparietal junction including the angular gyrus) or knowledge about the person (temporal pole). Frequently, this can also require the indirect involvement of technical reasoning to apprehend the physical dimension of the situation (e.g., it is raining, and two umbrellas can provide shelter from the rain), which can be crucial for inferring mental states (e.g., Alex uses one umbrella but does not give the other to Mary). For the mentalizing task, the participants were shown a comic strip conveying a short story and had to select between two additional cartoons the appropriate ending to that story (*Brunet et al., 2000*; *Völlm et al., 2006*). There were two experimental conditions in this task (*Figure 1D*): Reasoning only on the physical dimension of the event (PHYS-Only condition) and inferring an intention combined with reasoning on the physical dimension of the event (INT + PHYS condition). We predicted that the technical-reasoning network should be recruited in both conditions but the mentalizing network would be only involved in the INT + PHYS condition, allowing us to distinguish the cognitive processes implicated in the causal understanding of physical events versus events implying inferring an intention.

To sum up, the contribution that this study aims to make is to test the idea that technical reasoning might be implicated in the situations in which we need to think about the physical properties of our world. In line with previous work, we predicted that this specific form of technical cognition engages a network of brain areas, among which the area PF within the left supramarginal gyrus plays a central role.

## Results

### Behavioural results

All the behavioural results are given in *Figure 2*. As shown, scores were higher in the experimental conditions than for the control conditions for all the tasks (all p < 0.05). In other words, the experimental conditions were more difficult than the control conditions. This difference in terms of difficulty can also be illustrated by the fact that some participants performed at or below the chance level in the experimental conditions whereas none did so in the control conditions.

### Whole-brain results

All the activations described below are reported with their MNI coordinates in *Supplementary files 1–7*.

#### Generalizability of the technical-reasoning network

As explained above, we predicted an implication of the technical-reasoning network in the mechanical problem-solving task of Experiment 1 as well as in the psychotechnical task of Experiment 2. As shown in *Figure 3A*, the whole-brain analyses revealed that the mechanical problem-solving task engaged an almost all-left network of areas, comprising the left supramarginal gyrus within the inferior parietal lobe (including the area PF), the left IFG (opercular and triangular parts), and the left superior parietal cortex and dorsal premotor cortex. The psychotechnical task of Experiment 2 generated a more bilateral network, with greater activation in both supramarginal gyri (including the left area PF but not the right area PF), the opercular part of both IFG, both LOTC, and both superior parietal and dorsal premotor cortices (*Figure 3B*). Taken together, these results validate our prediction in indicating the

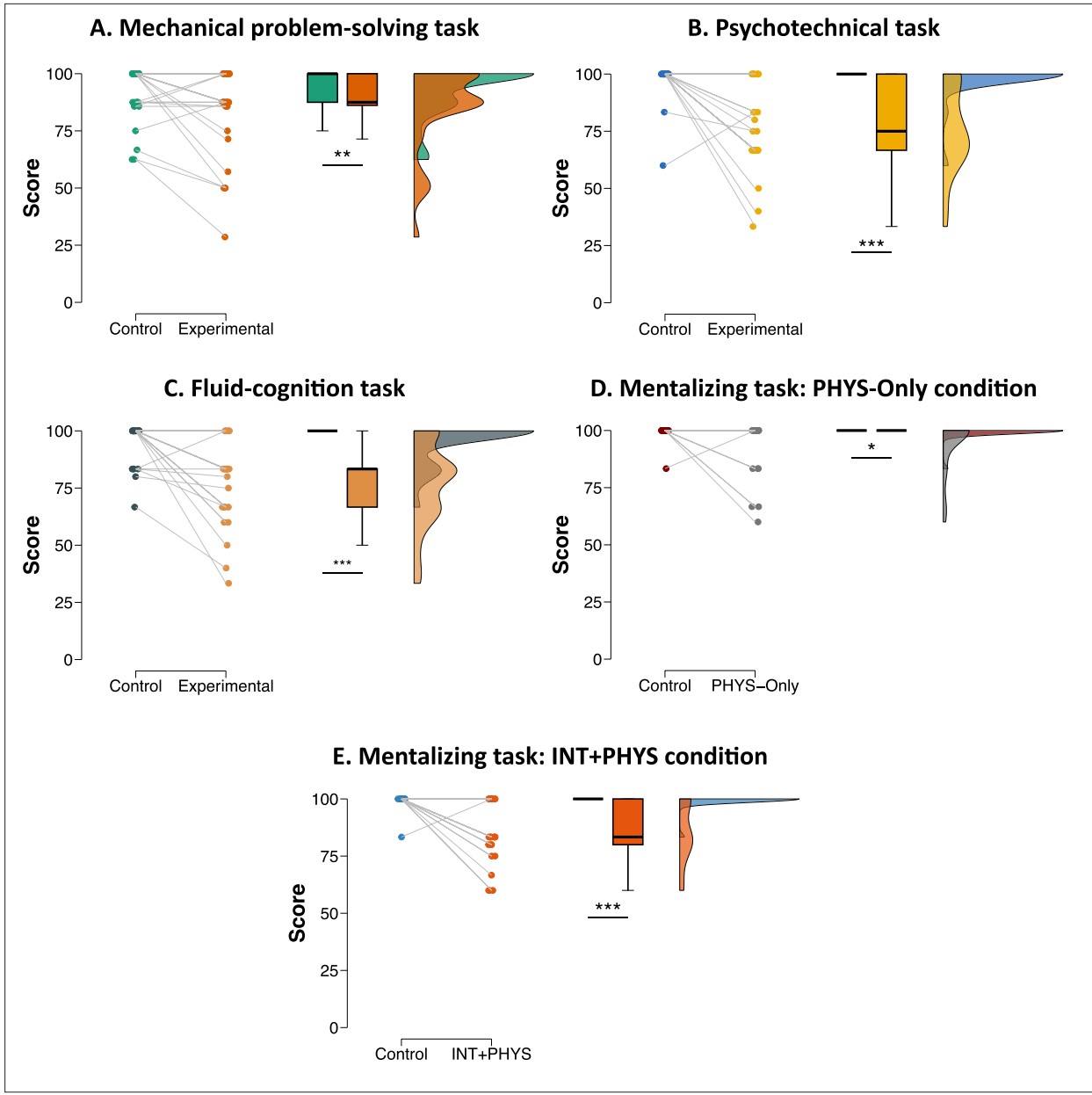

**Figure 2.** Behavioural results. The scores represent the percentage of correct responses. Boxplots indicate the upper quartile, median and lower quartile. *p < 0.05; **p < 0.01; ***p < 0.001.

involvement of the left area PF, which is central to the technical-reasoning network. This confirms that the technical-reasoning network depends upon the recruitment of the left area PF, even if additional cognitive processes involving other peripheral brain areas can be engaged depending on the task. This will be discussed in the final section of this article.

We also hypothesized that the technical-reasoning network should be recruited in the PHYS-Only and INT + PHYS conditions of the mentalizing task of Experiment 2. Before their extensive subsequent presentation, we will focus here on the results from these conditions allowing us to test our generaliz-ability hypothesis. The whole-brain analyses indicated preferential activation in the left supramarginal gyrus (including the left area PF) in both conditions (*Figure 3D, E*). Finally, the conjunction analysis of the four experimental conditions (the mechanical problem-solving task, the psychotechnical task, and the PHYS-Only and INT + PHYS conditions of the mentalizing task) led to the only activation in the left supramarginal gyrus (including the left area PF; *Figure 3G*). These findings confirmed that the

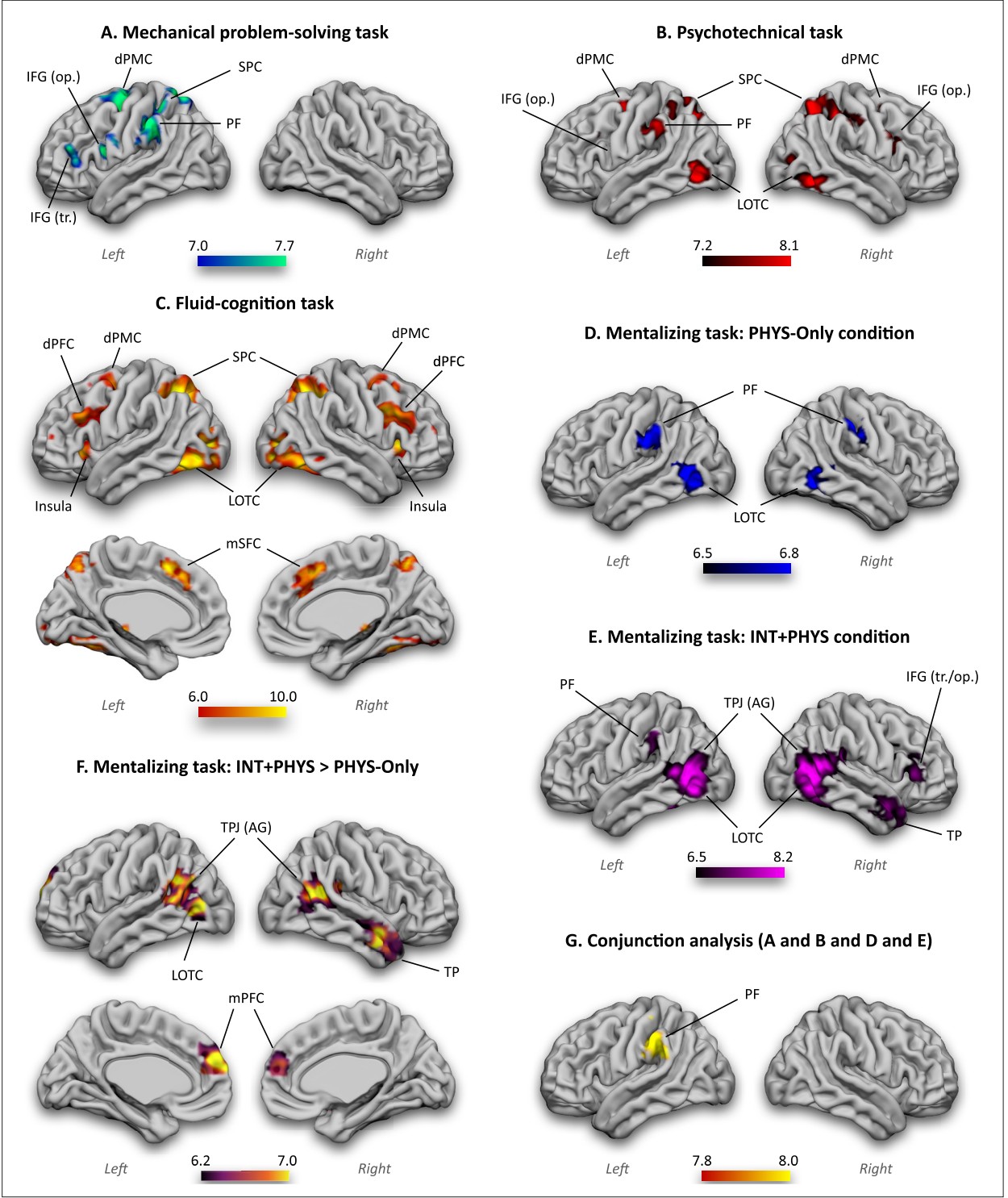

**Figure 3.** Whole-brain univariate results. The generalizability of the technical-reasoning network is supported by the activation of the left area PF in the mechanical problem-solving task (**A**), the psychotechnical task (**B**), and the PHYS-Only (**D**) and INT + PHYS (**E**) conditions of the mentalizing task. The conjunction analysis (**G**) also confirmed it. The specificity of the technical-reasoning network is also supported by the absence of activation of the left area PF in the fluid-cognition task (**C**) and in the contrast of the INT + PHYS condition to the PHYS-Only condition (**F**). In (**B**), IFG (op.) is indicated on the left hemisphere even if it is not visible on this view. Left, left hemisphere; right, right hemisphere; PF, parietal area F; IFG, inferior frontal gyrus (op., opercular part; tr., triangular part); dPMC; dorsal premotor cortex; SPC, superior parietal cortex; LOTC, lateral occipitotemporal cortex; dPFC, dorsal prefrontal cortex; TPJ, temporoparietal junction; AG, angular gyrus; TP, temporal pole; mPFC, medial prefrontal cortex. The colour bars represent the z-values.

left area PF, central to the technical-reasoning network, is recruited in any situation requiring physical understanding.

## Specificity of the technical-reasoning network

Even if technical reasoning is a non-verbal and causal form of reasoning, it must not be conflated with other non-verbal and/or causal forms of reasoning, such as fluid cognition and mentalizing. Therefore, we predicted that distinct cerebral networks are recruited for technical reasoning, fluid cognition, and mentalizing. The whole-brain results for the fluid-cognition task of Experiment 2 confirmed our prediction (*Figure 3C*). We found activation of both dorsal prefrontal cortices and medial superior frontal cortices, which are characteristic to fluid cognition but not to technical reasoning. Additional activation was reported in both LOTC, insulae, superior parietal cortices, and dorsal premotor cortices. No activation of the inferior parietal lobes was found. Concerning the mentalizing task of Experiment 2, the PHYS-Only condition revealed a bilateral network of areas, comprising both supramarginal gyri (including the right area PF and the left area PF as described above) and LOTC (*Figure 3D*). The INT + PHYS condition highlighted a bilateral network of areas comprising the left supramarginal gyrus (including the left area PF as described above), both LOTC, the right IFG (opercular and triangular parts), both temporoparietal junctions (including the angular gyri) and the right temporal pole (*Figure 3E*). Importantly, the contrast of the INT + PHYS condition to the PHYS-Only condition revealed a network of bilateral areas that characterize the mentalizing network, namely both temporoparietal junctions (including both angular gyri), both medial prefrontal cortices, and the right temporal pole (*Figure 3F*). No activation of brain areas of the technical-reasoning network survived this contrast.

## Region of interest results

We conducted additional analyses to test the robustness of our findings. One of our results was that we did not report any specific activation of the left area PF in the fluid-cognition task contrary to the mechanical problem-solving task, the psychotechnical task, and the PHYS-Only and INT + PHYS conditions of the mentalizing task. However, this negative result needed exploration at the region of interest (ROI) level. Therefore, we created a spherical ROI of the left area PF with a 5-mm radius in the MNI standard space (–59; –31; 40). This ROI was literature-defined to ensure the independence of its selection (*Reynaud et al., 2016*). ROI results are shown in *Figure 4*. The analyses confirmed the results obtained with the whole-brain analyses by indicating a greater activation of the left area PF in the mechanical problem-solving task, the psychotechnical task, and the PHYS-Only and INT + PHYS conditions of the mentalizing task (all p < 0.001), but not in the fluid-cognition task (p = 0.35). As mentioned above, the experimental conditions of all the tasks were more difficult than their control conditions. As a result, the specific activation of the left area PF documented above could simply reflect that this area responds to a greater extent in a harder condition relative to an easy condition of a task. This interpretation is nevertheless ruled out by the results obtained with the fluid-cognition task. We did not report a specific activation of the left area PF in this task while its experimental condition was more difficult than its control condition. To test more directly this effect of difficulty, we conducted new ROI analyses by removing all the participants who performed at or below 50% (*Figure 4—figure supplement 1*). These new analyses replicated the initial analyses by showing a greater activation of the left area PF in the mechanical problem-solving task, the psychotechnical task, and the PHYS-Only and INT + PHYS conditions of the mentalizing task (all p < 0.001), but not in the fluid-cognition task (p = 0.48). In sum, the ROI analyses corroborated the whole-brain analyses and ruled out the potential effect of difficulty.

The conjunction analysis reported was subject to at least two key limitations that needed to be overcome to assure a correct interpretation of our findings. The first was that the tasks could recruit the same regions for different cognition functions (same-region-but-different-function interpretation). The second was that the activated regions of the different tasks could be adjacent but did not overlap at finer resolutions (adjacency interpretation). We tested the same-region-but-different-function interpretation by conducting additional ROI analyses, which consisted of correlating the specific activation of the left area PF (i.e., difference in terms of mean Blood-Oxygen Level Dependent [BOLD] parameter estimates between the experimental condition minus the control condition) in the psychotechnical task, the fluid-cognition task, and the PHYS-Only and INT + PHYS conditions of the mentalizing task.

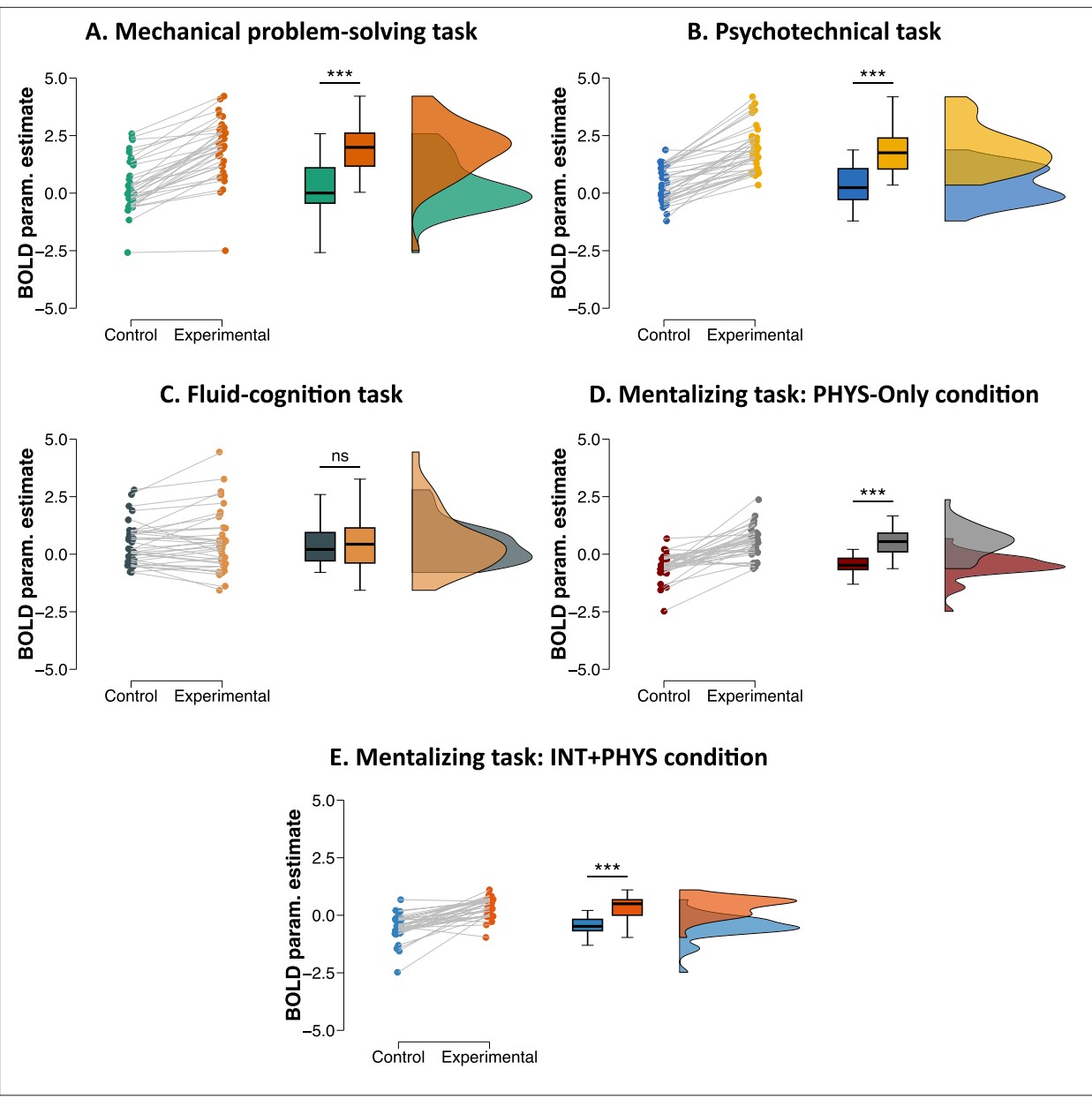

**Figure 4.** Region of interest (ROI) univariate results (left area PF). The results are shown here for (**A**) the mechanical problem-solving task, (**B**) the psychotechnical task, (**C**) the fluid-cognition task, and (**D**) the PHYS-Only and (**E**) INT + PHYS conditions of the mentalizing task. BOLD param. estimate refers to the mean BOLD activation value in the left area PF. Boxplots indicate the upper quartile, median and lower quartile. ns, not significant; ***p < 0.001.

The online version of this article includes the following figure supplement(s) for figure 4:

**Figure supplement 1.** Region of interest (ROI) univariate results (left area PF) for participants who performed at or above 50%.

This analysis did not include the mechanical problem-solving task because the sample of participants was not the same for this task. As shown in **Figure 5**, we found significant correlations between all the tasks that were hypothesized as recruiting technical reasoning, that is, the psychotechnical task and the PHYS-Only and INT + PHYS conditions of the mentalizing task (all p < 0.05). By contrast, no significant correlation was obtained between these three tasks and the fluid-cognition task (all p > 0.15). This finding invalidates the same-region-but-different-function interpretation by revealing a coherent pattern in the activation of the left area PF in situations in which participants were supposed to reason technically. We examined the adjacency interpretation by analysing the specific locations of individual peak activations within a 12-mm radius sphere centred on the left area PF ROI coordinates (−59; −31;

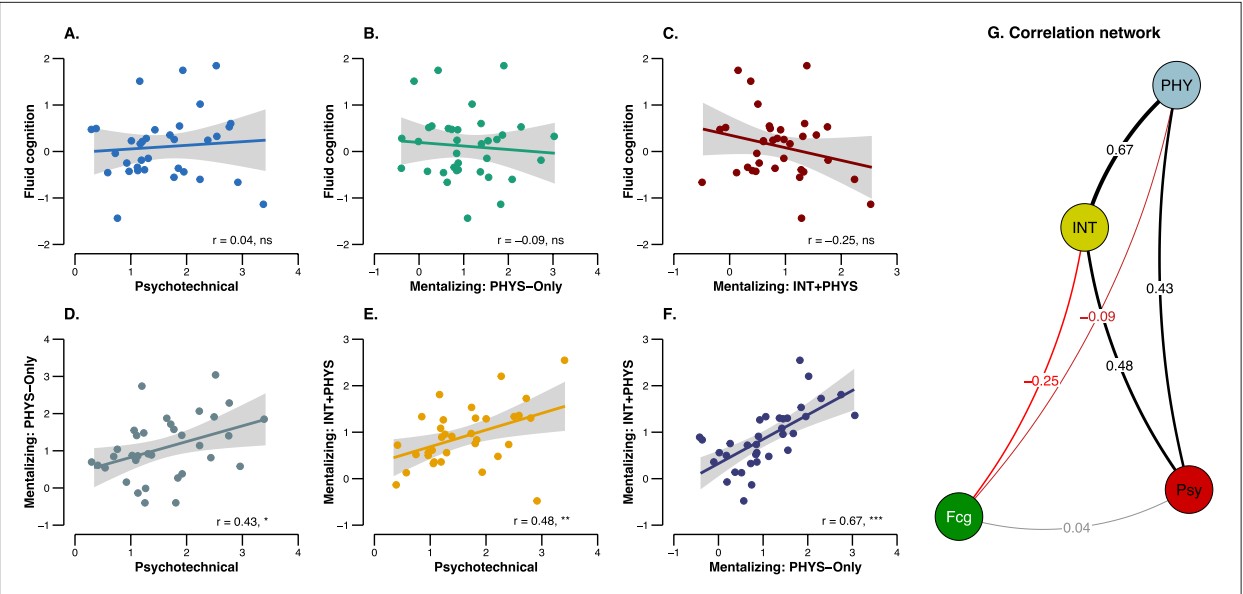

**Figure 5.** Correlations between mean BOLD parameter estimates of the left area PF in the different tasks. (**A-F**) shows the pairwise correlations between the psychotechnical task, the fluid-cognition task, and the PHYS-Only and (**E**) INT + PHYS conditions of the mentalizing task, and (**G**) the network that summarizes these pairwise correlations. The BOLD parameter estimates refer here to the difference in terms of mean BOLD activation between the experimental condition minus the control condition. The straight lines represent the linear model fits, and the light shaded areas are the standard errors. ns, not significant; *p < 0.05; **p < 0.01; ***p < 0.001; Psy, psychotechnical task; Fcg, fluid-cognition task; PHY, mentalizing: PHYS-Only condition; INT, mentalizing: INT + PHYS condition.

40) for each task and each subject. Results are reported in *Figure 6*. As can be seen, the peaks of the fluid-cognition task were located more anteriorly, in the left area PFt (Parietal Ft) and the postcentral cortex, compared to the peaks of the other four tasks, which were more posterior, in the left area PF. Statistical analyses based on the *y* coordinates of the individual activation peaks confirmed this description (*Figure 6*). Indeed, the *y* coordinates of the peaks of the mechanical problem-solving task, the psychotechnical task and the PHYS-Only and INT + PHYS conditions of the mentalizing task were posterior to the *y* coordinates of the peaks of the fluid-cognition task (all p < 0.05), whereas no significant differences were reported between the four tasks (all p > 0.05). These findings speak against the adjacency interpretation by revealing that participants recruited the same part of the left area PF to perform tasks involving technical reasoning.

## Discussion

Humans have created a wide range of technologies that have helped them colonize the whole surface of the Earth and beyond. Capitalizing on early Goldenberg's intuitions (*Goldenberg and Hagmann, 1998*; *Goldenberg and Spatt, 2009*), the technical-reasoning hypothesis assumes that this idiosyncratic technological trajectory reflects a specific form of technical cognition that involves a cerebral network in which the left area PF of the inferior parietal lobe is central (*Osiurak and Reynaud, 2020*; *Osiurak et al., 2023*). However, two characteristics of this reasoning remained to be tested, that is, its generalizable but specific nature. Here, we report two neuroimaging studies that confirmed these two characteristics. In the following lines, we discuss in turn the key findings of the present experiments, by stressing how they allow us to pave the way for future research on technical cognition.

The first key finding is that the left area PF, central to the technical-reasoning network, is systematically recruited in any situations involving physical events, confirming the generalizable dimension of technical reasoning. Neuropsychological evidence has indicated that damage to the left area PF impairs both common and novel tool use (*Goldenberg and Spatt, 2009*; *Martin et al., 2016*). Previous neuroimaging studies have also revealed that this brain area is specifically activated when (1) people focus on the mechanical actions between a tool and an object (*Reynaud et al., 2016*), (2) watch another individual use tools with objects (*Reynaud et al., 2019*), (3) reason about physical

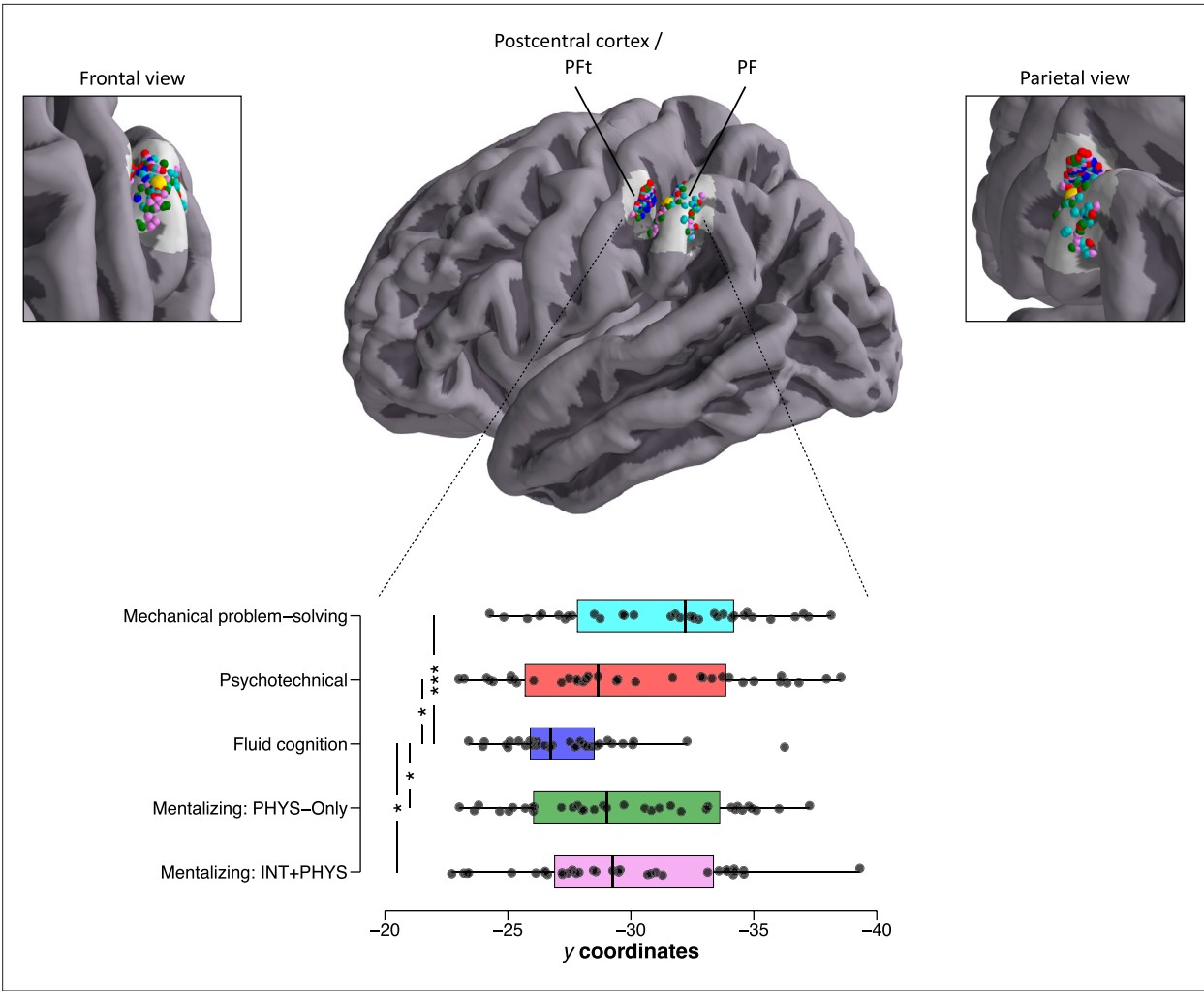

**Figure 6.** Individual *y* coordinates for the left area PF region of interest (ROI) as a function of the task. The individual maximum activation peaks of the mechanical problem-solving task, the psychotechnical task and the PHYS-Only and INT + PHYS conditions of the mentalizing task are located more posteriorly, in the area PF, than those of the fluid-cognition task, which are located more anteriorly in the area PFt and the postcentral cortex. The yellow sphere corresponded to the centre of the ROI (–59; –31; 40). Boxplots indicate the upper quartile, median and lower quartile. Only the significant comparisons are given. *p < 0.05; ***p < 0.001.

events (**Fischer et al., 2016**), or (4) look at physical events without being explicitly instructed to reason about them (**Pramod et al., 2022**). The present study corroborates these results in showing that this brain area is also preferentially recruited when people use novel tools to solve physical problems, or reason about physical events. More importantly, the conjunction analysis that included the mechanical problem-solving task, the psychotechnical task, and the PHYS-Only and INT + PHYS conditions of the mentalizing task confirmed this finding. Taken together, these findings suggest that the technical-reasoning network '*can provide the foundations for future research on this quintessentially human trait: Using, making, and reasoning about tools and more generally shaping the physical world to our ends*' (**Allen et al., 2020**; p. 29309).

It should be clear here that we do not advocate the localizationist position simply stating that activation in the left area PF is the necessary and sufficient condition for technical reasoning. We rather defend the view according to which it requires a network of interacting brain areas, one of them – and of major importance – being the left area PF. This allows the engagement of different configurations of cerebral areas in different technical-reasoning tasks, but with a central process acting as a stable component: The left area PF. Thus, when people intend to use physical tools, it can work in concert with brain regions specific to object manipulation and motor control, thereby forming another network, the tool-use network. It can also interact with brain regions specific to intentional gestures

to form a 'social-learning' network that allows people to enhance their understanding about the physical aspects of a technical task (e.g., the making of a tool) through communicative gestures such as pointing gestures (*Bluet et al., 2025*). The major challenge for future research is to specify the nature of the cognitive process supported by the left area PF and that might be involved in the broad understanding of the physical world. One possibility is that the left area PF might be a hub as is the temporal pole for semantic cognition (*Ralph et al., 2017*). Another is derived from the technical-reasoning hypothesis, which was initially developed from the observation of patients with tool-use disorders, who met difficulties in selecting and even making appropriate tools to solve mechanical problems. Although a link between problem-solving and reasoning can be drawn, it remains that associative learning is a viable candidate to explain how people solve mechanical problems or even more generally make predictions about physical events (*Heyes, 2023*). Finally, others have suggested the notion of an 'intuitive' physics engine, a kind of simulator able to perform physical inference (*Battaglia et al., 2013*; *Yildirim et al., 2019*; *Schwettmann et al., 2019*; *Fischer and Mahon, 2021*). Future research is needed to explore these alternative interpretations and potentially reconcile some of them, which is the essential step for the development of a field dedicated to the study of technical cognition.

The second key finding is that the technical-reasoning network is not recruited in other non-verbal and/or causal reasoning forms, confirming that it is a specific form of reasoning oriented towards the physical world. Indeed, no activation of the left area PF was found for the fluid-cognition task, differentiating the logical-reasoning process engaged for solving this task from technical reasoning. This is in line with previous results dissociating technical reasoning from fluid cognition based on behavioural experiments. For instance, in a recent study (*De Oliveira et al., 2019*), the technical-reasoning skills and fluid-cognition skills of 245 participants were assessed with tasks close to those employed here for assessing these two forms of reasoning. A factor analysis with varimax rotation was conducted and corroborated the orthogonality of technical-reasoning and fluid-cognition tasks. At the cerebral level, previous studies have indicated the key role of the prefrontal cortex in fluid cognition (*Hobeika et al., 2016*), which is also found in the present results. However, we did not report any engagement of the technical-reasoning network in the fluid-cognition task, confirming that technical reasoning is a non-verbal form of reasoning specific to the physical domain that must be distinguished from fluid cognition. The results of the mentalizing task are also informative in this respect. In the INT + PHYS condition of this task, the attribution of the characters' intentions was accompanied by the mandatory understanding of the physical event associated. On the contrary, the PHYS-Only condition only implied reasoning about the physical actions performed by the characters of the cartoon. Our results confirmed this 'hierarchy' of processes. The left area PF was found for the two conditions as stated previously, but the INT +PHYS condition engaged an additional network of areas previously associated with mentalizing skills (*Gallagher and Frith, 2003*; *Molenberghs et al., 2016*; *Schurz et al., 2021*): The bilateral temporoparietal junctions, the bilateral medial prefrontal cortex, and the right temporal pole. In broad terms, the results of the mentalizing task indicate that causal reasoning has distinct forms and that it recruits distinct networks of the human brain (Social domain: Mentalizing; Physical domain: Technical reasoning), which can nevertheless interact together to solve day-to-day problems in which several domains are involved, such as in the INT + PHYS condition of the mentalizing task.

The third key finding is that the technical-reasoning network involves brain areas whose recruitment varies according to the task. Understanding the role played by these additional brain areas is crucial not only to describe the flexibility of the technical-reasoning network but also to continue to clarify (by exclusion) the role of the left area PF. The first additional area to consider is the left IFG, particularly its opercular part. This area has been repeatedly found in previous research on technical reasoning. We reported activation of left IFG in the mechanical problem-solving task (–50; 7; 20) and the psychotechnical task (–50; 7; 27). Even if the left IFG activation did not survive corrections in the conjunction analysis, it must be noted that a cluster of a small size ($k = 68$) was located around the peak at (–48; 9; 19). These peaks were situated in the close vicinity of the ones reported in two meta-analyses related to tool-use understanding (–49; 8; 31) (*Reynaud et al., 2016*) and tool-use observation (–51; 5; 33) (*Reynaud et al., 2019*). The functional role of the left IFG in the context of tool use has been previously discussed (*Reynaud et al., 2019*) and a plausible hypothesis is that the left IFG integrates the multiple constraints posed by the physical situation to set the ground for a correct reasoning process, such as it could be involved in syntactic language processing (for a somewhat similar view, see *Buxbaum and Randerath, 2018*). It has already been shown that language and tool

use share syntactic processes in the basal ganglia of the brain (*Thibault et al., 2021*). This hypothesis of IFG as a 'constraint combiner' for language and tool use is currently under investigation in our group and needs solid experimental proof.

Another additional area that has been found in three experimental conditions (psychotechnical task, and the PHYS-Only and INT + PHYS conditions of the mentalizing task) was the LOTC. These specific activations could be related to familiarity with the tools or objects shown (*Vingerhoets, 2008*; *Lesourd et al., 2021*). Indeed, we did not find the activation of this brain area in the mechanical problem-solving task, which focused on unfamiliar tool-use situations. The three others experimental conditions included common tools such as hammers or screwdrivers for the psychotechnical task, and common objects used as tools for the mentalizing tasks (e.g., stick, rope, and chair). Thus, the activation of the LOTC might reflect the involvement of semantic processes whose involvement would remain to understand.

Finally, activation was also found in the right area PF for the PHYS-Only condition of the mentalizing task. A recent morphometry study (*Federico et al., 2022*) showed that the cortical thickness of the left area PF predicts technical-reasoning and visuospatial performance, whereas the right area PF only predicts visuospatial performance, confirming the distinction between these two abilities (*Mitko and Fischer, 2020*). These findings suggest that the right area PF is recruited along with the left area PF when the task makes high demands on both technical and visuospatial dimensions, as in the PHYS-Only condition of the mentalizing task. This interpretation, although viable, remains unlikely given that there is no reason to consider that this condition makes heavier demands on both technical and visuospatial dimensions than the mechanical problem-solving task or the psychotechnical task. In broad terms, the role played by the right area PF in the technical-reasoning network remains an open issue.

Before concluding, we would like to point out two potential limitations of the present study. The first limitation concerns the fact that the literature has documented the recruitment of the left area PF in many neuroimaging experiments in which there was no need to reason about physical events (e.g., language tasks). This can be easily illustrated by entering the left area PF coordinates in the Neurosynth database. This finding could be enough to refute the idea that this brain area is specific to technical reasoning. Although this limitation deserves to be recognized, it is also true for many other findings. For instance, sensory or motor brain regions such as the precentral or the postcentral cortex have been found activated in many non-motor tasks, the visual word form area in non-language tasks, or the Heschl's gyrus in non-musical tasks. This remains a major challenge for scientists, the question being how to solve these inconsistencies that can result from statistical errors or stress that considerable effort is needed to understand the very functional nature of these brain areas. Thus, understanding that the left area PF is central to physical understanding can be viewed as a first essential step before discovering its fundamental function, as suggested by the functional polyhedral approach (*Genon et al., 2018*). The second limitation concerns the alternative interpretation that the left area PF is not central to technical reasoning but to the storage of sensorimotor programs about the prototypical manipulation of common tools. Here, we show that the left area PF is recruited even in situations in which participants do not have to process common manipulable tools. For instance, some items of the psychotechnical task consisted of pictures of tractor, boat, pulley, or cannon. The fact that we found a common activation of the left area PF in such tasks as well as in the mechanical problem-solving task, in which participants could nevertheless simulate the motor actions of manipulating novel tools, indicates that this brain area is not central to tool manipulation but to physical understanding. That being said, some may suggest that viewing a boat or a cannon is enough to incite the simulation of motor actions, so our tasks were not equipped to distinguish between the manipulation- and the reasoning-based approach. We have already shown that the left area PF is more involved in tasks that focus on the mechanical dimension of the tool-use action (e.g., the mechanical interaction between a tool and an object) than its motor dimension (i.e., the interaction between the tool and the effector [e.g., *Reynaud et al., 2019*; *Reynaud et al., 2016*]). Nevertheless, we recognize that future research is still needed to test the predictions derived from these two approaches.

Following the effort undertaken by others (*Bayani et al., 2021*; *Gärdenfors and Högberg, 2017*; *Gärdenfors, 2021*; *Strachan et al., 2021*; *Charbonneau et al., 2024*; *Sperber, 1996*; *Sperber and Hirschfeld, 2004*), the present study contributes to integrating cognitive science into the cultural evolution field in which technical cognition – if not cognition (*Heyes, 2016*; *Heyes, 2018*) – has remained peripheral to the debate on the origins and evolution of human technology. As *Wynn et al.,*

*2017* stated, '[e]ven archaeologists, for whom technical remains are the primary data source, have tended to privilege language and symbol use in discussion of the modern mind' (p. 21). Yet, recent accounts have proposed that non-social-cognitive skills such as causal understanding or technical reasoning might have played a crucial role in cumulative technological culture (*Whiten, 2022*; *Osiurak et al., 2023*; *Vale et al., 2021*). Support for these accounts comes from micro-society experiments, which have demonstrated that the improvement of technology over generations is accompanied by an increase in its understanding (*Osiurak et al., 2021*; *Osiurak et al., 2022*), or that learners' technical-reasoning skills are a good predictor of cumulative performance in such micro-societies (*De Oliveira et al., 2019*; *Osiurak et al., 2016*). While behavioural experiments tend to demonstrate the impact of technical reasoning on cumulative technological culture, the present findings offer a neural reality to these behavioural results and inspire new questions: Which, if any, cognitive subcomponents of technical reasoning are specific to the human species? Can cumulative technological culture emerge without technical reasoning? How to distinguish technical reasoning from associative learning in humans (*Heyes, 2023*)? How do humans translate their physical understanding into explanations? All these and other fascinating questions constitute a research agenda for investigating the co-evolution of the human brain, cognition, and technology.

## Materials and methods

### Participants

The study was conducted in the Laboratory for the Study of Cognitive Mechanisms at the University of Lyon (Lyon, France) and in the Lyon Neuroimaging Department (CERMEP, Lyon, France). For both experiments, participants were randomly recruited through advertisements posted on social media websites. One week before the MRI session, the participants signed the informed consent to take part in the study and were seen by a medical doctor to ascertain their eligibility for the neuroimaging session. All the participants were right-handed, had a normal or corrected-to-normal vision, provided informed consent, and reported no history of neurological or psychiatric disorder. All the participants signed written consent and were given a monetary incentive for their time (60€). The study was in line with the Declaration of Helsinki and was approved by a French Ethics Committee (N°ID-RCB: 2018-A00734-51). Thirty-four participants ($M_{age}$ = 24.21, SD = 4.04, range: 18–36; female gender: $n$ = 20, male gender: $n$ = 14) took part in Experiment 1 and 35 new participants ($M_{age}$ = 24.31, SD = 5.37, range: 20–44; female gender: $n$ = 23, male gender: $n$ = 12) in Experiment 2. Inclusion in the final sample required that head motion during scanning did not exceed 0.5 mm displacement (i.e., framewise displacement) between consecutive volumes on 90% of volumes. No participants were excluded based on this criterion.

### Stimuli and design

For all experiments, the participants were thoroughly briefed on the instructions for completing the tasks just before the scanning session. Two practice trials per condition were proposed but did not reappear inside the scanner. The first experiment consisted of a $T_1$-weighted anatomical scan and a functional run for the mechanical problem-solving task. The other experiment was scanned in a single session with two functional runs separated with a $T_1$-weighted anatomical scan. The psychotechnical and mentalizing tasks were part of the first run, and the fluid-cognition task was scanned in the second run. We used a within-subject design with blocks of different lengths for each condition for the Experiment 1, and a fixed length of 30 s for all the other experiments emanating from the presentation of, respectively, 5, 3, and 3 images for, respectively, 6, 10, and 10 s each, for the psychotechnical, fluid-cognition and mentalizing tasks. Blocks were separated with a fixation cross for 15 s in all conditions. Participants' answers were collected via a button box held in their right hand.

#### Mechanical problem-solving task

Eight experimental and eight control blocks were either 25 s (4 blocks), 32.5 s (2 blocks), or 40 s (2 blocks) long. Each block consisted of 3, 4, or 5 trials. Each trial was composed as follows: The image of the 3D glass box only was displayed for 4 s. In the control condition, a black square mask was applied on the picture. Then, for the next 3 s, two tools (experimental condition) or two missing pieces (control condition) were added on the left and right sides of the 3D box (*Figure 1*, *Figure 1—figure*

*supplement 1*). Before the scanning session, the participants were informed that, in the experimental condition, five distinct tools could be used to solve the mechanical problems (*Figure 1A*), which consisted in moving a small red cubic element trapped in the 3D glass box from its original location into a new target location. In the scanner, the participants had to figure out for 4 s and for each mechanical problem how to solve it by using one of the tools shown before the scanning session. Then, two tools were presented for 3 s, and they had to decide which was the correct one to solve the mechanical problem by pressing either the left or the right button of the button box. The control condition was a visual completion task. The participants scrutinized the 3D glass box for 4 s and then had 3 s to decide which of the two missing pieces presented was the correct one to fill the mask. A 500-ms fixation cross separated the trials. On the last trial of each block, a red frame appeared around the visual scene, signalling to the participants that their response was awaited. Consequently, 3 additional seconds were added to the display of the box with tools or pieces on the sides to allow for motor response. Blocks alternated between the experimental condition and the control condition. The items of this task are available at https://osf.io/hfrmu/.

## Psychotechnical task

Two images were shown simultaneously for 6 s, in blocks of five boards. On the last board of each block, a red frame instructed the participants to physically answer within 2 s by pressing either the left or the right button of the button box, for motor responses. Blocks alternated between the experimental condition and the control condition, six times each. The participants had to select which of the two presented situations was the correct one or the most effective one in the experimental condition (*Figure 1B*), whereas, in the control condition, they had to select the situation containing a square (*Figure 1—figure supplement 1B*). The items were adapted from the NV5 (https://www.pearson-clinical.fr/nv5r) and NV7 (https://www.pearsonclinical.fr/nv7) batteries. The adaptations consisted in reducing the number of options from 4 to 2 and in modifying one part of the picture to create a square (for the control condition). As these batteries are commercialized, we did not provide the items – even the modified ones – in an open-access repository. Nevertheless, the items can be available on request. As the original items of the NV5 and NV7 batteries, the items used in the study were in black and white. The pictures shown in *Figure 1*, *Figure 1—figure supplement 1* are nevertheless deliberately in colour so as to move further away from the original items.

## Fluid-cognition task

Boards were shown for 10 s, in blocks of three boards. For the last board of each block, a red frame reminded the participants to answer by pressing either the left or the right button of the button box in the scanner during the last 2 s of the board presentation. Blocks alternated between the experimental condition and the control condition, six times each. The experimental condition required fluid reasoning (*Figure 1C*) whereas the control condition required only visuospatial pattern completion (*Figure 1—figure supplement 1C*). The items were adapted from the Raven's Progressive Matrices test (https://www.pearsonclinical.fr/pm-progressive-matrices-de-raven). The adaptations consisted in presenting two lines of three options, one on the left and the other on the right of the screen. The participants had to select the line with the correct option. As this test is commercialized, we did not provide the items – even the modified ones – in an open-access repository. Nevertheless, the items can be available on request. As the original items of the Raven's Progressive Matrices test, the items used in the study were in black and white. The pictures shown in *Figure 1*, *Figure 1—figure supplement 1* are nevertheless deliberately in colour so as to move further away from the original items.

## Mentalizing task

For each condition, blocks of three boards were constituted, and each board was presented in two different steps. First, the superior part of the board was shown for 6 s, for the participants to try to make sense of the cartoon first. Then the bottom part was presented for 4 additional seconds, with the top part remaining on display. On the last image of each block, a red frame appeared, indicating to the participants that a physical answer via the button box was required, during the last 2 s of the presentation. Blocks were therefore 30 s long and were repeated six times each. Half of the cartoons in each condition involved a single character, the other half more than one character (all but one implied two characters, and one implied a character versus a crowd). For the two experimental conditions

(i.e., INT + PHYS and PHYS-Only conditions; *Figure 1D*), the participants had to choose the cartoon with the probable ending to the story depicted in the three first drawings, and for the control condition, they had to select which cartoon was already present in the first three ones (*Figure 1—figure supplement 1D*). In the PHYS-Only condition, the selection only needed to understand the physical context. In the INT + PHYS condition, the selection needed to understand both the physical context and the social context. The items were adapted from the task used by *Völlm et al., 2006*. The main adaptations concerned the control condition, which was not present in *Völlm et al., 2006*. Indeed, in their study, the control conditions were PHYS-Only conditions. Birgit Völlm gave us the permission to make available the items of the task, which can be found at https://osf.io/hfrmu/. Note that the items available at this open-access repository do not correspond to all the items used in the task originally developed by *Völlm et al., 2006* but only to the items used in the present study.

## fMRI data acquisition

For both experiments, 'neuroimaging data were acquired on a 3T Siemens Prisma Scanner (Siemens, Erlangen, Germany) using a 64-channel head coil. BOLD images were recorded with $T_2$*-weighted echo-planar images (EPI) acquired with the multi-band sequence. Functional images were all collected as oblique-axial scans aligned with the anterior commissure–posterior commissure (AC–PC) line with the following parameters' (p. 6529; *Lesourd et al., 2023*): 1030 (mechanical problem-solving task), 1000 (psychotechnical and fluid-cognition tasks), 763 (mentalizing task) volumes per run, 57 slices, TR/TE = 1400/30 ms, flip angle = 70°, field of view = 96 × 96 mm$^2$, slice thickness = 2.3 mm, voxel size = 2.3 × 2.3 × 2.3 mm$^3$, multi-band factor = 2. Structural $T_1$-weighted images were collected using an MPRAGE sequence (224 sagittal slices, TR/TE = 3000/2.93 ms, inversion time = 1100 ms, flip angle = 8°, 224 × 256 mm FOV, slice thickness = 0.8 mm, voxel size = 0.8 × 0.8 × 0.8 mm$^3$).

## Preprocessing of fMRI data

For both experiments, '[s]tructural $T_1$-weighted images were segmented into tissue type (GM: grey matter; WM: white matter; CSF: cerebrospinal fluid tissues) using the Computational Anatomy Toolbox (CAT12; http://dbm.neuro.uni-jena.de/cat12/) segmentation tool, in order to facilitate the normalization step. Functional data were analysed using SPM12 (Wellcome Department of Cognitive Neurology, http://www.fil.ion.ucl.ac.uk/spm) implemented in MATLAB (Mathworks, Sherborn, MA)' (p. 6529; *Lesourd et al., 2023*). Four steps were followed for the preprocessing for univariate analyses: (1) Realignment to the mean EPI image with 6-head motion correction parameters and unwarping using the FieldMap toolbox from SPM12; (2) 'co-registration of the individual functional and anatomical images; (3) normalization towards MNI template; and (4) spatial smoothing of functional images (Gaussian kernel with 5 mm FWHM)' (p. 6530; *Lesourd et al., 2023*).

## Group analysis

A general linear model was created using design matrices containing one regressor (explanatory variable) for each condition (i.e., mechanical problem-solving task and its control condition for Experiment 1, and psychotechnical, fluid-cognition and mentalizing tasks and their respective control conditions for Experiment 2) modelled as a boxcar function (with onsets and durations corresponding to the start of each stimulus of that condition) convolved with the canonical hemodynamic response function as well as its temporal and derivatives dispersion. Regressors of non-interest resulting from 3D head motion estimation (*x*, *y*, *z* translation and three axes of rotation) and a set of cosine regressors for high-pass filtering were added to the design matrix. The model was estimated for each participant, also considering the average signal in the run. After model estimation, we computed contrasts at the first level (i.e., experimental conditions versus control conditions) and then transferred to a second-level group analysis (one-sample *t*-test) to obtain the brain regions more activated in experimental than on the control condition, for the four tasks. We present results maps with a significance threshold set at p < 0.05 with family-wise error (FWE) correction at the cluster level unless stated otherwise. The maps were thresholded at a minimal size of *k* = 120 voxels per cluster.

## Conjunction analysis

We performed a conjunction analysis using statistical parametric maps testing for the conjunction-null hypothesis with the maximum p-value statistic over the four contrasts for the mechanical

problem-solving task, the psychotechnical task, and the INT + PHYS and PHYS-ONLY conditions of the mentalizing task. A first p-value map was computed by intersecting the three contrasts from the psychotechnical task and the two mentalizing tasks as repeated measures on the same participants. The resulting uncorrected *T*-map from the conjunction-null analysis ran into SPM12 was then transformed into a p-value map with the appropriate degrees of freedom. Then, in a second step, the uncorrected *T*-map from the mechanical problem-solving task was transformed into a p-value map, taking into account the number of participants minus 1 as degrees of freedom. The two p-value maps were in a third step intersected with a conjunction ran as the maximum p-value over the two p-value maps, allowing to test for the conjunction-null hypothesis and to infer a conjunction of $k = 4$ effects at significant voxels. The resulting p-value map was then thresholded at the level of $p < 0.05$ (FWE corrected) and for a minimum size of $k = 100$ voxels per cluster.

### ROI analyses

We created a spherical ROI of the left area PF with a 5-mm radius in the MNI standard space (–59; –31; 40). This ROI was literature-defined to ensure the independence of its selection (*Reynaud et al., 2016*). Fo each task, we used regression modelling in R (*R Development Core Team, 2011*) (lmerTest package *Kuznetsova et al., 2017*) to fit a linear model with 'mean BOLD parameter estimate' as outcome variable, 'condition' (Experimental versus Control) as fixed effect, and 'participant's identity' as random effect. For correlational analyses, we computed the difference in terms of mean BOLD parameter estimate between the experimental condition minus the control condition of each task and used Pearson correlation coefficients. For the *y* coordinates analyses, peak activations were identified by determining the maximum value coordinates within a 12-mm radius sphere centred on the left area PF ROI coordinates (–59; –31; 40) for each task and each subject. To investigate the spatial distribution of these peaks, we focused specifically on their *y* coordinates and fitted a linear model with '*y* coordinate' as outcome variable, 'task Mechanical problem-solving versus Psychotechnical versus Fluid-cognition versus PHYS-Only condition of the mentalizing task versus INT + PHYS condition of the mentalizing task' as fixed effect, and 'participant's identity' as random effect. Post hoc pairwise *t*-tests were computed with a Holm–Bonferroni correction. Independent *t*-tests were used to compare the mechanical problem-solving task to the other four tasks as the samples of participants were different. Paired *t*-tests were performed for the comparisons between the other tasks. Statistical significance was set at $p < 0.05$.

### Behavioural analyses

For each task, we used regression modelling in R to fit a linear model with 'score as outcome variable, 'condition' (Experimental versus Control) as fixed effect, and 'participant's identity' as random effect. Statistical significance was set at $p < 0.05$.

### Code availability

Codes used in this study are available at https://osf.io/hfrmu/.

### Materials availability

Materials used in this study are available at https://osf.io/hfrmu/.

## Acknowledgements

We warmly thank Emanuelle Reynaud for her precious help in many aspects of the study. We also thank Emmanuel De Oliveira, Boris Alexandre, and Alexandrine Faye for their help in designing the experimental stimuli. We thank Birgit Völlm for giving us the permission to reproduce the items of her tests and to make them available in an open-access repository. This work was supported by grants from the French National Research Agency (ANR; Project TECHNITION: ANR-21-CE28-0023-01; FO, YR, and ML) and the Région Auvergne-Rhône-Alpes (NUMERICOG-2017-900-EA 3082 EMC-R-2075; FO).

# Additional information

## Funding

| Funder | Grant reference number | Author |
|---|---|---|
| Agence Nationale de la Recherche | ANR-21-CE28-0023-01 | François Osiurak<br>Yves Rossetti<br>Mathieu Lesourd |
| Région Auvergne-Rhône-Alpes | NUMERICOG-2017-900-EA 3082 EMC-R-2075 | François Osiurak |

The funders had no role in study design, data collection, and interpretation, or the decision to submit the work for publication.

## Author contributions

François Osiurak, Conceptualization, Data curation, Supervision, Funding acquisition, Investigation, Visualization, Methodology, Writing – original draft, Project administration, Writing – review and editing; Giovanni Federico, Methodology, Writing – review and editing; Arnaud Fournel, Formal analysis, Methodology, Writing – review and editing; Vivien Gaujoux, Investigation, Writing – review and editing; Franck Lamberton, Danièle Ibarrola, Resources, Data curation, Investigation, Writing – review and editing; Yves Rossetti, Conceptualization, Funding acquisition, Investigation, Writing – review and editing; Mathieu Lesourd, Conceptualization, Data curation, Formal analysis, Supervision, Funding acquisition, Visualization, Methodology, Writing – original draft, Writing – review and editing

## Author ORCIDs

François Osiurak (ID) https://orcid.org/0000-0003-3449-6377
Giovanni Federico (ID) http://orcid.org/0000-0001-8395-0770
Arnaud Fournel (ID) http://orcid.org/0000-0002-4344-1987
Vivien Gaujoux (ID) http://orcid.org/0000-0003-2486-0454
Franck Lamberton (ID) http://orcid.org/0000-0002-4655-4397
Danièle Ibarrola (ID) http://orcid.org/0000-0001-8999-8740
Yves Rossetti (ID) https://orcid.org/0000-0001-8867-4496
Mathieu Lesourd (ID) https://orcid.org/0000-0002-1011-3047

## Ethics

The study was in line with the Declaration of Helsinki and was approved by a French Ethics Committee (N°ID-RCB: 2018-A00734-51).

Reviewer #1 (Public review): https://doi.org/10.7554/eLife.94578.3.sa1
Reviewer #2 (Public review): https://doi.org/10.7554/eLife.94578.3.sa2
Reviewer #3 (Public review): https://doi.org/10.7554/eLife.94578.3.sa3
Author response https://doi.org/10.7554/eLife.94578.3.sa4

# Additional files

## Supplementary files

Supplementary file 1. Local maxima of activation clusters (MNI stereotactic coordinates) for the mechanical problem-solving task (Experimental condition > Control condition).

Supplementary file 2. Local maxima of activation clusters (MNI stereotactic coordinates) for the psychotechnical task (Experimental condition > Control condition).

Supplementary file 3. Local maxima of activation clusters (MNI stereotactic coordinates) for the fluid-cognition task (Experimental condition > Control condition).

Supplementary file 4. Local maxima of activation clusters (MNI stereotactic coordinates) for the mentalizing task (PHYS-Only condition > Control condition).

Supplementary file 5. Local maxima of activation clusters (MNI stereotactic coordinates) for the mentalizing task (INT + PHYS condition > Control condition).

Supplementary file 6. Local maxima of activation clusters (MNI stereotactic coordinates) for the

mentalizing task (INT + PHYS condition > PHYS-Only condition).

Supplementary file 7. Local maxima of activation clusters (MNI stereotactic coordinates) for the conjunction analysis (mechanical problem-solving AND psychotechnical AND INT + PHYS AND PHYS-Only).

MDAR checklist

## Data availability

The codes and data used in this study are available at https://osf.io/hfrmu/.

The following dataset was generated:

| Author(s) | Year | Dataset title | Dataset URL | Database and Identifier |
|---|---|---|---|---|
| Osiurak F, Fournel A | 2023 | The left PF technical-cognition area | https://osf.io/hfrmu/ | Open Science Framework, hfrmu |

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
