## [Editor Report · eLife Assessment]

This **valuable** study used functional MRI experiments to identify the involvement of a left parietal area (PF) in reasoning about the physical properties of actions, objects, and events. **Solid** evidence was shown regarding the commonalities and differences across different types of reasoning tasks, yet the methodological and theoretical interpretations require further scrutiny. The study would be of interest to researchers studying the cognitive and neural mechanisms of reasoning and problem solving.

---

## [Referee Report · Reviewer #1 (Public review)]

In this study, Osiurak and colleagues investigate the neurocognitive basis of technical reasoning. They use multiple tasks from two neuroimaging studies to show that the area PF is central to technical reasoning and plays an essential role in tool-use and non-tool-use physical problem-solving, as well as both conditions of mentalizing tasks. They also demonstrate the specificity of technical reasoning, finding that area PF is not involved in the fluid-cognition task or the mentalizing network (INT+PHYS vs. PHYS-only). This work enhances our understanding of the neurocognitive basis of technical reasoning that supports advanced technologies.

Strengths:

- The topic this study focuses on is intriguing and can help us understand the neurocognitive processes involved in technical reasoning and advanced technologies.

- The researchers collected fMRI data from multiple tasks. The data is rich and encompasses the mechanical problem-solving task, psychotechnical task, fluid-cognition task, and mentalizing tasks.

- The article is well written.

The authors have addressed many of the reviewers' concerns in their response. They utilized both correlation analysis and coordinate analysis to tackle alternative hypotheses, namely the same-region-but-different-function interpretation and the adjacency interpretation. Additionally, ROI analysis was conducted to validate the negative results. These additional analyses have enhanced the reliability of the findings. This study offers valuable insights into the neurocognitive mechanisms underlying technical reasoning.

Weaknesses:

While the authors attempted to address the limitations of overlap analysis by correlating activation across different tasks within subjects, this issue could not be entirely resolved due to the constraints of the current experimental design. The mechanical problem-solving task was not included since the sample of subjects differed from that of other tasks. Furthermore, the fluid-cognition task was not scanned in the same run as the psychotechnical and mentalizing tasks, which may have contributed to a lack of correlation between them, thereby affecting result interpretation. Moreover, the core cognitive focus of this study, technical reasoning, may be influenced by assumptions about motion-related information. While this issue has been discussed in the discussion section, further evidence is needed to substantiate this interpretation.

---

## [Referee Report · Reviewer #2 (Public review)]

Strengths:

The authors have done a nice job providing additional data in response to reviewer feedback. I appreciate that accuracy plots are now included, as well as a separate analysis where differences in parameter estimates are performed for participants whose accuracy data were above chance levels. I also appreciate the new figure with the sphere ROIs for each participant, as they help us appreciate anatomical variability in the peak response separately for each task.

I have four concerns related to the weaknesses of the study:

(1) Although the results still hold when removing participants whose accuracy was 50% or less, a major limitation of this study is that participants made a button press response only to the last trial in a block. This is problematic because a participant could get all trials in a block correct except for the last one, or a participant could get all trials in a block wrong, and performance would be considered equivalent-as a consequence, it is not possible for one to know if participants who are at chance are performing differently from participants who are not at chance, and it is not possible to control for variance in reaction time (a concern also raised by reviewer 3).

(2) My second concern relates to the way in which the data are interpreted based on thresholding. There is above-threshold activation in the left SMG for all tasks except the fluid cognition task. The z-scores associated with significant voxels in Figure 3 are very strong (minimum z is 6). If one were to relax the threshold of the group level maps to, e.g., p < .001, uncorrected, FDR q < .05, or FWER of .10, there will be overlapping voxels outside the SMG. The discussion of the left SMG in the manuscript is prominent and narrowly construed-the left SMG is discussed as if it were 'the' region: "This confirms that the technical-reasoning network depends upon the recruitment of the left area PF, even if additional cognitive processes involving other peripheral brain areas can be engaged depending on the task" (pp. 9). My intuition is there will be numerous other areas of overlap when using a threshold that is still highly significant (e.g., z = 3 or 4). So, for proponents of the technical reasoning hypothesis, is there a counterfactual or alternative brain area/network/system not in the left SMG?

(3) I like the new Figure 6 because it shows variability in the location of the peak coordinate at the level of single participants. And, indeed, there's considerable variability that is typical when localizing ROIs in single participants. My concern is the level at which hypothesis testing is performed. An independent SMG ROI is used to extract parameter estimates and correlate responses between tasks to show a pattern of correlation that comports with a technical reasoning model of left SMG function. This is a fine approach but it does not rule out the so-called 'same region different function' interpretation because it relies on correlation-one cannot reverse infer that the left SMG is carrying out the same function across different tasks because the response in that area is more strongly correlated between certain tasks. This finding points to that possibility and makes interesting predictions for future studies to pursue, but it cannot tell us whether common functions in the left SMG are involved in each task. E.g., one interesting prediction for future studies is to test if patients with lesions to this site are disproportionately more inaccurate in the experimental condition of the mechanical problem solving task, the psychotechnical task, the mentalizing task, but not the fluid cognition task.

(4) I appreciated the approach to testing the adjacency interpretation by showing the sphere and peak Y coordinate across the tasks. It is interesting that across the groups, there is no difference in the peak Y coordinate of the psychotechnical task and both conditions of the mentalizing task, whereas the peak Y coordinate in the fluid intelligence task is more anterior in the post-central gyrus across participants (why is that?). But why restrict the analysis to just the Y coordinate? A rigorous way to test the adjacency hypothesis is to compute Euclidean distance among X, Y, and Z coordinates between any two tasks collected in the same participant. One can then test if the Euclidean distance between, e.g., the psychotechnical task and one condition of the mentalizing task is smaller than the Euclidean distance between the psychotechnical task and the fluid cognition task. Similarly, one can test whether Euclidean distance between the INT and PHY conditions of the mentalizing task is smaller than the Euclidean distance between the INT and psychotechnical task or PHY and psychotechnical task. There is no justification to restrict this analysis to the anterior-posterior dimension only.

---

## [Referee Report · Reviewer #3 (Public review)]

The authors have responded very thoughtfully to many of the points raised, and the revised manuscript will make a useful contribution to our understanding of some of the computations performed by area PF. In particular, the investigators' addition of analyses of peak activations, their additional clarifications that area PF is likely to be part of a larger network concerned with technical reasoning, and their responses to the reviewers' concerns about differential task difficulty have strengthened the conclusions that can be drawn from the study.

The authors' response does not completely mitigate the concern noted by all 3 reviewers that the control tasks were easier than their corresponding experimental tasks (for everything but the fluid cognition task). The specific trouble with this issue can be appreciated by looking at Figure 4A, for example, which shows that area PF was activated for many individuals in both the control task and the experimental mechanical problem-solving task, but more so for the latter. Since the experimental task was harder (and more trial time was likely spent on task trying to solve it), the concern remains that area PF was driven harder by the experimental task in part due to the more challenging nature of that task.

The revised manuscript counters that the fluid cognition task was also harder than its control condition, yet did not activate PF more than its control condition. But this response seems to sidestep the central point of the reviewers' concerns: the fundamental computations that underlie the technical reasoning tasks may also be present in the respective (non technical-reasoning-based) control tasks and drive area PF activations to greater or lesser degrees based on how much they tax those computations. The fact that the fluid cognition experimental task and control task are not differentially difficult does not mitigate this concern, it just suggests that neither of those tasks tap the same fundamental computations, whatever they may be. (As an added note, Figures 2 and 4 show that both the PHYS-only and INT+PHYS mentalizing tasks only weakly activated PF, and both of these tasks were easier than the other technical cognition tasks).

The new ROI analysis with removal of subjects who performed below 50% in the revised manuscript is somewhat helpful, but there are two remaining issues: (1) chance performance is defined by a binomial test in this case, so scores somewhat above 50% may still be at chance depending on the number of items, and thus there may have been subjects who were not removed who could not perform the tasks; (2) it would have been convincing to include accuracy as a covariate in the modeling of BOLD parameter estimates for the remaining above-chance subjects to ensure that all reported effects remain once differential task difficulty is taken into account. It also appears that the legend for Figure S2, which indicates that the figure includes just subjects who performed at or below 50%, may not be correct; does the figure instead show data from subjects who performed at or above 50%?

Despite these remaining concerns, there are many aspects of this revised study that render it a useful contribution that will likely spur further research in this very interesting area.

---

## [Author Response]

The following is the authors’ response to the original reviews

**Reviewer #1 (Public Review):**
Summary:In this study, Osiurak and colleagues investigate the neurocognitive basis of technical reasoning. They use multiple tasks from two neuroimaging studies and overlap analysis to show that the area PF is central for reasoning, and plays an essential role in tool-use and non-tool-use physical problem-solving, as well as both conditions of mentalizing task. They also demonstrate the specificity of the technical reasoning and find that the area PF is not involved in the fluid-cognition task or the mentalizing network (INT+PHYS vs. PHYS-only). This work suggests an understanding of the neurocognitive basis of technical reasoning that supports advanced technologies.Strengths:-The topic this study focuses on is intriguing and can help us understand the neurocognitive processes involved in technical reasoning and advanced technologies.-The researchers obtained fMRI data from multiple tasks. The data is rich and encompasses the mechanical problem-solving task, psychotechnical task, fluid-cognition task, and mentalizing task.-The article is well written.

We sincerely thank Reviewer 1 for their positive and very helpful comments, which helped us improve the MS. Thank you.

Weaknesses:- Limitations of the overlap analysis method: there are multiple reasons why two tasks might activate the same brain regions. For instance, the two tasks might share cognitive mechanisms, the activated regions of the two tasks might be adjacent but not overlapping at finer resolutions, or the tasks might recruit the same regions for different cognition functions.Thus, although overlap analysis can provide valuable information, it also has limitations.Further analyses that capture the common cognitive components of activation across different

tasks are warranted, such as correlating the activation across different tasks within subjects for a region of interest (i.e. the PF).

We thank Reviewer 1 for this comment. We added new analyses to address the two alternative interpretations stressed here by Reviewer 1, namely, the same-region-but-differentfonction interpretation and the adjacency interpretation. The new analyses ruled out both alternative interpretations, thereby reinforcing our interpretation.

“The conjunction analysis reported was subject to at least two key limitations that needed to be overcome to assure a correct interpretation of our findings. The first was that the tasks could recruit the same regions for different cognition functions (same-region-but-different-function interpretation). The second was that the activated regions of the different tasks could be adjacent but did not overlap at finer resolutions (adjacency interpretation). We tested the same-region-but-different-function interpretation by conducting additional ROI analyses, which consisted of correlating the specific activation of the left area PF (i.e., difference in terms of mean Blood-Oxygen Level Dependent [BOLD] parameter estimates between the experimental condition minus the control condition) in the psychotechnical task, the fluid-cognition task, and the PHYS-Only and INT+PHYS conditions of the mentalizing task. This analysis did not include the mechanical problem-solving task because the sample of participants was not the same for this task. As shown in Fig. 5, we found significant correlations between all the tasks that were hypothesized as recruiting technical reasoning, i.e., the psychotechnical task and the PHYS-Only and INT+PHYS conditions of the mentalizing task (all *p* < .05). By contrast, no significant correlation was obtained between these three tasks and the fluid-cognition task (all *p* > .15). This finding invalidates the same-region-but-different-function interpretation by revealing a coherent pattern in the activation of the left area PF in situations in which participants were supposed to reason technically. We examined the adjacency interpretation by analysing the specific locations of individual peak activations within the left area PF ROI for the mechanical problemsolving task, the psychotechnical task, the fluid-cognition task, and the PHYS-Only and INT+PHYS conditions of the mentalizing task. These peaks, which corresponded to the maximum value of activation obtained for each participant within the left area PF ROI, are reported in Fig. 6. As can be seen, the peaks of the fluid-cognition task were located more anteriorly, in the left area PFt (Parietal Ft) and the postcentral cortex, compared to the peaks of the other four tasks, which were more posterior, in the left area PF. Statistical analyses based on the *y* coordinates of the individual activation peaks confirmed this description (Fig. 6). Indeed, the *y* coordinates of the peaks of the mechanical problem-solving task, the psychotechnical task and the PHYS-Only and INT+PHYS conditions of the mentalizing task were posterior to the *y* coordinates of the peaks of the fluid-cognition task (all *p* < .05), whereas no significant differences were reported between the four tasks (all *p* > .05). These findings speak against the adjacency interpretation by revealing that participants recruited the same part of the left area PF to perform tasks involving technical reasoning.” (p. 11-13)

Control tasks may be inadequate: the tasks may involve other factors, such as motor/ actionrelated information. For the psychotechnical task, fluid-cognition task, and mentalizing task, the experiment tasks need not only care about technical-cognition information but also motor-related information, whereas the control tasks do not need to consider motor-related information (mainly visual shape information). Additionally, there may be no difference in motor-related information between the conditions of the fluid-cognition task. Therefore, the regions of interest may be sensitive to motor-related information, affecting the research conclusion.

We thank Reviewer 1 for this comment. We added a specific section in the discussion that addresses this limitation.

“The second limitation concerns the alternative interpretation that the left area PF is not central to technical reasoning but to the storage of sensorimotor programs about the prototypical manipulation of common tools. Here we show that the left area PF is recruited even in situations in which participants do not have to process common manipulable tools. For instance, some items of the psychotechnical task consisted of pictures of tractor, boat, pulley, or cannon. The fact that we found a common activation of the left area PF in such tasks as well as in the mechanical problem-solving task, in which participants could nevertheless simulate the motor actions of manipulating novel tools, indicates that this brain area is not central to tool manipulation but to physical understanding. That being said, some may suggest that viewing a boat or a cannon is enough to incite the simulation of motor actions, so our tasks were not equipped to distinguish between the manipulation-based approach and the reasoning-based approach. We have already shown that the left area PF is more involved in tasks that focus on the mechanical dimension of the tool-use action (e.g., the mechanical interaction between a tool and an object) than its motor dimension (i.e., the interaction between the tool and the effector [e.g., 24, 40]). Nevertheless, we recognize that future research is still needed to test the predictions derived from these two approaches.” (p. 18-19)

-Negative results require further validation: the cognitive results for the fluid-cognition task in the study may need more refinement. For instance, when performing ROI analysis, are there any differences between the conditions? Bayesian statistics might also be helpful to account for the negative results.

We agree that our negative results required further validation. We conducted the ROI analyses suggested by Reviewer 1, which confirmed the initial whole-brain analyses.

“Region of interest (ROI) results. We conducted additional analyses to test the robustness of our findings. One of our results was that we did not report any specific activation of the left area PF in the fluid-cognition task contrary to the mechanical problem-solving task, the psychotechnical task, and the PHYS-Only and INT+PHYS conditions of the mentalizing task. However, this negative result needed exploration at the ROI level. Therefore, we created a spherical ROI of the left area PF with a radius of 12 mm in the MNI standard space (–59; –31; 40). This ROI was literature-defined to ensure the independence of its selection (40). ROI results are shown in Fig. 4. The analyses confirmed the results obtained with the whole-brain analyses by indicating a greater activation of the left area PF in the mechanical problem-solving task, the psychotechnical task, and the PHYS-Only and INT+PHYS conditions of the mentalizing task (all *p* < .001), but not in the fluid-cognition task (*p* = .35).” (p. 10-11)

**Reviewer #1 (Recommendations For The Authors):**
(1) I may not fully grasp some of the arguments. In the abstract, what does the term "intermediate-level" mean, and why is it an intermediate-level state? In the sentence "the existence of a specific cognitive module in the human brain dedicated to materiality", I cannot see a clear link between technical cognition and the word "materiality".

We used the term materiality to refer to a potential human trait that allows us to shape the physical world according to our ends, by using, making tools and transmiting them to others. This is a reference to Allen et al. (2020; PNAS): “We hope this empirical domain and modeling framework can provide the foundations for future research on this quintessentially human trait: using, making, and reasoning about tools and more generally shaping the physical world to our ends” (p. 29309). Scientists (including archaeologists, economists, psychologists, neuroscientists) interested in human materiality have tended to focus on how we manipulate things according to our thought (motor cognition) or how we conceptualize our behaviour to transmit it to others (language, social cognition). However, little has been said on the intermediate level, that is, technical cognition. We added the term “technical cognition” here, which should help to make the connection more quickly.

“Yet, little has been said about the intermediate-level cognitive processes that are directly involved in mastering this materiality, that is, technical cognition.” (p. 2)

(2) The introduction could provide more details on why the issue of "generalizability and specificity" is important to address, to clarify the significance of the research question.

We followed this comment and added a sentence to explain why it is important to address this research question. Again, we thank Reviewer 1 for their helpful comments.

“Here we focus on two key aspects of the technical-reasoning hypothesis that remain to be addressed: Generalizability and specificity. If technical reasoning is a specific form of reasoning oriented towards the physical world, then it should be implicated in *all* (the generalizability question) *and only* (the specificity question) the situations in which we need to think about the physical properties of our world.” (p. 5)

**Reviewer #2 (Public Review):**
Summary:The goal of this project was to test the hypothesis that a common neuroanatomic substrate in the left inferior parietal lobule (area PF) underlies reasoning about the physical properties of actions and objects. Four functional MRI (fMRI) experiments were created to test this hypothesis. Group contrast maps were then obtained for each task, and overlap among the tasks was computed at the voxel level. The principal finding is that the left PF exhibited differentially greater BOLD response in tasks requiring participants to reason about the physical properties of actions and objects (referred to as technical reasoning). In contrast, there was no differential BOLD response in the left PF when participants engaged in fMRI variant of the Raven's progressive matrices to assess fluid cognition.Strengths:This is a well-written manuscript that builds from extensive prior work from this group mapping the brain areas and cognitive mechanisms underlying object manipulation, technical reasoning, and problem-solving. Major strengths of this manuscript include the use of control conditions to demonstrate there are differentially greater BOLD responses in area PF over and above the baseline condition of each task. Another strength is the demonstration that area PF is not responsive in tasks assessing fluid cognition - e.g., it may just be that PF responds to a greater extent in a harder condition relative to an easy condition of a task. The analysis of data from Task 3 rules out this alternative interpretation. The methods and analysis are sufficiently written for others to replicate the study, and the materials and code for data analysis are publicly available.

We sincerely thank Reviewer 2 for their precious comments, which helped us improve the MS.

Weaknesses:The first weakness is that the conclusions of the manuscript rely on there being overlap among group-level contrast maps presented in Figure 2. The problem with this conclusion is that different participants engaged in different tasks. Never is an analysis performed to demonstrate that the PF region identified in e.g., participant 1 in Task 2 is the same PF region identified in Participant 1 in Task 4.

We added new analyses that demonstrated that “the PF region identified in e.g., participant 1 in Task 2 is the same PF region identified in Participant 1 in Task 4”. We thank Reviewer 2 for this comment, because these new analyses reinforced our interpretation.

“The conjunction analysis reported was subject to at least two key limitations that needed to be overcome to assure a correct interpretation of our findings. The first was that the tasks could recruit the same regions for different cognition functions (same-region-but-different-function interpretation). The second was that the activated regions of the different tasks could be adjacent but did not overlap at finer resolutions (adjacency interpretation). We tested the same-region-but-different-function interpretation by conducting additional ROI analyses, which consisted of correlating the specific activation of the left area PF (i.e., difference in terms of mean Blood-Oxygen Level Dependent [BOLD] parameter estimates between the experimental condition minus the control condition) in the psychotechnical task, the fluid-cognition task, and the PHYS-Only and INT+PHYS conditions of the mentalizing task. This analysis did not include the mechanical problem-solving task because the sample of participants was not the same for this task. As shown in Fig. 5, we found significant correlations between all the tasks that were hypothesized as recruiting technical reasoning, i.e., the psychotechnical task and the PHYS-Only and INT+PHYS conditions of the mentalizing task (all *p* < .05). By contrast, no significant correlation was obtained between these three tasks and the fluid-cognition task (all *p* > .15). This finding invalidates the same-region-but-different-function interpretation by revealing a coherent pattern in the activation of the left area PF in situations in which participants were supposed to reason technically. We examined the adjacency interpretation by analysing the specific locations of individual peak activations within the left area PF ROI for the mechanical problemsolving task, the psychotechnical task, the fluid-cognition task, and the PHYS-Only and INT+PHYS conditions of the mentalizing task. These peaks, which corresponded to the maximum value of activation obtained for each participant within the left area PF ROI, are reported in Fig. 6. As can be seen, the peaks of the fluid-cognition task were located more anteriorly, in the left area PFt (Parietal Ft) and the postcentral cortex, compared to the peaks of the other four tasks, which were more posterior, in the left area PF. Statistical analyses based on the *y* coordinates of the individual activation peaks confirmed this description (Fig. 6). Indeed, the *y* coordinates of the peaks of the mechanical problem-solving task, the psychotechnical task and the PHYS-Only and INT+PHYS conditions of the mentalizing task were posterior to the *y* coordinates of the peaks of the fluid-cognition task (all *p* < .05), whereas no significant differences were reported between the four tasks (all *p* > .05). These findings speak against the adjacency interpretation by revealing that participants recruited the same part of the left area PF to perform tasks involving technical reasoning.” (p. 11-13)

A second weakness is that there is a variance in accuracy between tasks that are not addressed. It is clear from the plots in the supplemental materials that some participants score below chance (~ 50%). This means that half (or more) of the fMRI trials of some participants are incorrect. The methods section does not mention how inaccurate trials were handled. Moreover, if 50% is chance, it suggests that some participants did not understand task instructions and were systematically selecting the incorrect item.

It is true that the experimental conditions were more difficult than the control conditions, with some participants who performed at or below 50% in the experimental conditions. We added a section in the MS to stress this aspect. To examine whether this potential difficulty effect biased our interpretation, we conducted new ROI analyses by removing all the participants who performed at or below the chance level. These analyses revealed the same results as when no participant was excluded, suggesting that this did not bias our interpretation.

“As mentioned above, the experimental conditions of all the tasks were more difficult than their control conditions. As a result, the specific activation of the left area PF documented above could simply reflect that this area responds to a greater extent in a harder condition relative to an easy condition of a task. This interpretation is nevertheless ruled out by the results obtained with the fluid-cognition task. We did not report a specific activation of the left area PF in this task while its experimental condition was more difficult than its control condition. To test more directly this effect of difficulty, we conducted new ROI analyses by removing all the participants who performed at or below 50% (Fig. S2). These new analyses replicated the initial analyses by showing a greater activation of the left area PF in the mechanical problem-solving task, the psychotechnical task, and the PHYS-Only and INT+PHYS conditions of the mentalizing task (all *p* < .001), but not in the fluid-cognition task (*p* = .48). In sum, the ROI analyses corroborated the wholebrain analyses and ruled out the potential effect of difficulty.” (p. 11)

A third weakness is related to the fluid cognition task. In the fMRI task developed here, the participant must press a left or right button to select between 2 rows of 3 stimuli while only one of the 3 stimuli is the correct target. This means that within a 10-second window, the participant must identify the pattern in the 3x3 grid and then separately discriminate among 6 possible shapes to find the matching stimulus. This is a hard task that is qualitatively different from the other tasks in terms of the content being manipulated and the time constraints.

We acknowledge that the fluid-cognition task involved a design that differed from the other tasks. However, this was also true for the other tasks, as the design also differed between the mechanical problem-solving task, the psychotechnical task, and the mentalizing task. Nevertheless, despite these distinctions, we found a consistent activation of the left area PF in these tasks with different designs including in the psychotechnical task, which seemed as difficult as the fluid-cognition task.

“Region of interest (ROI) results. We conducted additional analyses to test the robustness of our findings. One of our results was that we did not report any specific activation of the left area PF in the fluid-cognition task contrary to the mechanical problem-solving task, the psychotechnical task, and the PHYS-Only and INT+PHYS conditions of the mentalizing task. However, this negative result needed exploration at the ROI level. Therefore, we created a spherical ROI of the left area PF with a radius of 12 mm in the MNI standard space (–59; –31; 40). This ROI was literature-defined to ensure the independence of its selection (40). ROI results are shown in Fig. 4. The analyses confirmed the results obtained with the whole-brain analyses by indicating a greater activation of the left area PF in the mechanical problem-solving task, the psychotechnical task, and the PHYS-Only and INT+PHYS conditions of the mentalizing task (all *p* < .001), but not in the fluid-cognition task (*p* = .35).” (p. 10-11)

In sum, this is an interesting study that tests a neuro-cognitive model whereby the left PF forms a key node in a network of brain regions supporting technical reasoning for tool and non-tool-based tasks. Localizing area PF at the level of single participants and managing variance in accuracy is critically important before testing the proposed hypotheses.

We thank Reviewer 2 for this positive evaluation and their suggestions. As detailed in our response, our revision took into consideration both the localization of the left area PF at the level of single participants and the variance in accuracy.

**Reviewer #2 (Recommendations For The Authors):**
Did the fMRI data undergo high-pass temporal filtering prior to modeling the effects of interest? Participants engaged in a long (17-24 minutes) run of fMRI data collection. Highpass filtering of the data is critically important when managing temporal autocorrelation in the fMRI response (e.g., see Shinn et al., 2023, Functional brain networks reflect spatial and temporal autocorrelation. Nature Neuroscience).

Yes. We added this information.

“Regressors of non-interest resulting from 3D head motion estimation (*x*, *y*, *z* translation and three axes of rotation) and a set of cosine regressors for high-pass filtering were added to the design matrix.” (p. 25-26)

Including scales in Figure 2 would help the reader interpret the magnitude of the BOLD effects.

We added this information in Figure 3 (Figure 2 in the initial version of the MS).

It was difficult to inspect the small thumbnail images of the task stimuli in Figure 1. Higher resolution versions of those stimuli would help facilitate understanding of the task design and trial structure.

We changed both Figure 1 and Figure S1.

**Reviewer #3 (Public Review):**
Summary:This manuscript reports two neuroimaging experiments assessing commonalities and differences in activation loci across mechanical problem-solving, technical reasoning, fluid cognition, and "mentalizing" tasks. Each task includes a control task. Conjunction analyses are performed to identify regions in common across tasks. As Area PF (a part of the supramarginal gyrus of the inferior parietal lobe) is involved across 3 of the 4 tasks, the investigators claim that it is the hub of technical cognition.Strengths:The aim of finding commonalities and differences across related problem-solving tasks is a useful and interesting one.The experimental tasks themselves appear relatively well-thought-out, aside from the concern that they are differentially difficult.The imaging pipeline appears appropriate.

We thank Reviewer 3 for their constructive comments, which helped us improve the MS.

Weaknesses:(1) MethodologicalAs indicated in the supplementary tables and figures, the experimental tasks employed differ markedly in (1) difficulty and (2) experimental trial time. Response latencies are not reported (but are of additional concern given the variance in difficulty). There is concern that at least some of the differences in activation patterns across tasks are the result of these fundamental differences in how hard various brain regions have to work to solve the tasks and/or how much of the trial epoch is actually consumed by "on-task" behavior. These difficulty issues should be controlled for by (1) separating correct and incorrect trials, and (2) for correct trials, entering response latency as a regressor in the Generalized Linear Models, (3) entering trial duration in the GLMs.

We thank Reviewer 3 for this comment. It is true that the experimental conditions were more difficult than the control conditions, with some participants who performed at or below 50% in the experimental conditions. We added a section in the MS to stress this aspect. We could not conduct new analyses by separating correct and incorrect trials because, for each task, participants had to respond only on the last item of the block. Therefore, we did not record a response for each event. Nevertheless, we could examine whether this potential difficulty effect biased our interpretation, by conducting new ROI analyses in which we removed all the participants who performed at or below the chance level. These analyses revealed the same results as when no participant was excluded, suggesting that this did not bias our interpretation.

“As mentioned above, the experimental conditions of all the tasks were more difficult than their control conditions. As a result, the specific activation of the left area PF documented above could simply reflect that this area responds to a greater extent in a harder condition relative to an easy condition of a task. This interpretation is nevertheless ruled out by the results obtained with the fluid-cognition task. We did not report a specific activation of the left area PF in this task while its experimental condition was more difficult than its control condition. To test more directly this effect of difficulty, we conducted new ROI analyses by removing all the participants who performed at or below 50% (Fig. S2). These new analyses replicated the initial analyses by showing a greater activation of the left area PF in the mechanical problem-solving task, the psychotechnical task, and the PHYS-Only and INT+PHYS conditions of the mentalizing task (all *p* < .001), but not in the fluid-cognition task (*p* = .48). In sum, the ROI analyses corroborated the wholebrain analyses and ruled out the potential effect of difficulty.” (p. 11)

A related concern is that the control tasks also differ markedly in the degree to which they were easier and faster than their corresponding experimental task. Thus, some of the control tasks seem to control much better for difficulty and time on task than others. For example, the control task for the psychotechnical task simply requires the indication of which array contains a simple square shape (i.e., it is much easier than the psychotechnical task), whereas the control task for mechanical problem-solving requires mentally fitting a shape into a design, much like solving a jigsaw puzzle (i.e., it is only slightly easier than the experimental task).

It is true that some control conditions could be easier than other ones. These differences reinforced the common activation found in the left area PF in the tasks hypothesized as involving technical reasoning, because this activation survived irrespective of the differences in terms of experimental design. For us, the rationale is the same as for a meta-analysis, in which we try to find what is common to a great variety of tasks. The only detrimental consequence we identified here is that this difference explained why we did not report a specific activation of the left area PF in the fluid-cognition task, as if the left area PF was more responsive when the task was difficult. This possibility assumes that the experimental condition of the fluid-cognition task is much more difficult than its control condition compared to what can be seen in the other tasks. As Reviewer 2 stressed in Point 1, this interpretation is unlikely, because the differences between the experimental and control conditions were similar to the fluid-cognition task in the mechanical problem-solving and psychotechnical tasks. In addition, again, the new ROI analyses in which we removed all the participants who performed at or below the chance level in expetimental conditions reproduced our initital results.

(2) TheoreticalThe investigators seem to overlook prior research that does not support their perspective and their writing seems to lack scientific objectivity in places. At times they over-reach in the claims that can be made based on the present data. Some claims need to be revised/softened.

As this comment is also mentioned below, please find our response to it below.

**Reviewer #3 (Recommendations For The Authors):**
(1) Because of the high level of detail, Figures 1 and S2 (particularly the mentalizing task and mechanical problem-solving task, and their controls) are very hard to parse, even when examined relatively closely. It is suggested that these figures be broken down into separate panels for Experiment 1 and Experiment 2 to facilitate understanding.

We changed both Figure 1 and Figure S1.

(2) The behavioral data (including response latencies) should be reported in the main results section of the paper and not in a supplement.

The behavioural data are now reported in the main results. We did not report response latencies because participants were not prompted to respond as quickly as possible.

“Behavioural results. All the behavioural results are given in Fig. 2. As shown, scores were higher in the experimental conditions than for the control conditions for all the tasks (all *p* < .05). In other words, the experimental conditions were more difficult than the control conditions. This difference in terms of difficulty can also be illustrated by the fact that some participants performed at or below the chance level in the experimental conditions whereas none did so in the control conditions.” (p. 8)

(3) The investigators seem to overlook prior research that does not support their perspective and their writing seems to lack scientific objectivity in places. At times they over-reach in the claims that can be made based on the present data. For example, claims that need to be revised/softened include:Abstract: "Area PF... can work along with social-cognitive skills to resolve day-to-day interactions that combine social and physical constraints". This statement is overly speculative.

This statement is based on the fact that we reported a combined activation of the technical-reasoning network and the mentalizing network in the INT+PHYS condition of the mentalizing task. This suggests that both networks need to work together for solving a day-today problem in which both the physical constraints of the situation and the intention of the individual must be integrated. Our findings replicated previous ones with a similar task (e.g., Brunet et al. 2000; Völlm et al., 2006), in which the authors gave an interpretation similar to ours in considering that this task requires understanding physical and social causes. Perhaps that the reference to the results of the mentalizing task was not explicit enough. We added “dayto-day” before “problem” in the part of the discussion in which we discuss this possibility to make this aspect clearer.

“In broad terms, the results of the mentalizing task indicate that causal reasoning has distinct forms and that it recruits distinct networks of the human brain (Social domain: Mentalizing; Physical domain: Technical reasoning), which can nevertheless interact together to solve day-to-day problems in which several domains are involved, such as in the INT+PHYS condition of the mentalizing task.” (p. 16)

Introduction: "The manipulation-based approach... remains silent on the more general cognitive mechanisms...that must also encompass the use of unfamiliar or novel tools". This statement seems to be based on an overly selective literature review. There are a number of studies in which the relationship between a novel and familiar tool selection/use has been explored (e.g., Buchman & Randerath, 2017; Mizelle & Wheaton, 2010; Silveri & Ciccarelli, 2009; Stoll, Finkel et al., 2022; Foerster, 2023; Foerster, Borghi, & Goslin, 2020; Seidel, Rijntjes et al., 2023).

We thank Reviewer 3 for this comment. Even if we accept the idea that we possess specific sensorimotor programs about tool manipulation, it remains that these programs cannot explain how an individual decides to bend a wire to make a hook or to pour water in a recipient to retrieve a target. As a matter of fact, such behaviour has been reported in nonhuman animals, such as crows (Weir et al., 2002, Nature) or orangutans (Mendes et al., 2007, Biology Letters). In these studies, the question is whether these nonhuman animals understand the physical causes or not, but the question of sensorimotor programs is never addressed (to our knowledge). This is also true in developmental studies on tool use (e.g., Beck et al., 2011, Cognition; Cutting et al., 2011, Journal of Experimental Child Psychology). This is what we meant here, that is, the manipulation-based approach is not equipped to explain how people solve physical problems by using or making tools – or any object – or by building constructions or producing technical innovations. However, we agree that some papers have been interested in exploring the link between common and novel tool use and have suggested that both could recruit common sensorimotor programs. It is noteworthy that these studies do not test the predictions from the manipulation-based approach versus the reasoning-based approach, so both interpretations are generally viable as stressed by Seidel et al. (2023), one of the papers recommended by Reviewer 3.

“Apparently, the presentation of a graspable object that is recognizable as a tool is sufficient to provoke SMG activation, whether one tends to see the function of SMG to be either “technical reasoning” (Osiurak and Badets 2016; Reynaud et al. 2016; Lesourd et al. 2018; Reynaud et al. 2019) or “manipulation knowledge” (Sakreida et al. 2016; Buxbaum 2017; Garcea et al. 2019b).” (Seidel et al., 2023; p. 9)

Regardless, as suggested by Reviewer 3, these papers deserve to be cited and this part needed to be rewritten to insist on the “making, construction, and innovation” dimension more than on the “unfamiliar and novel tool use” dimension to avoid any ambiguity.

“This manipulation-based approach has provided interesting insights (12–16) and even elegant attempts to explain how these sensorimotor programs could support the use of both unfamiliar or novel tools (17–20), but remains silent on the more general cognitive mechanisms behind human technology that include the use of common and unfamiliar or novel tools but must also encompass tool making, construction behaviour, technical innovations, and transmission of technical content.” (p. 3)

Introduction: "Here we focus on two important questions... to promote the technicalreasoning hypothesis as a comprehensive cognitive framework..."(italics added). This and other similar statements should be rewritten as testable scientific hypotheses rather than implying that the point of the research is to promote the investigators' preferred view.

We agree that our phrasing could seem inappropriate here. What we meant here is that the technical-reasoning hypothesis could become an interesting framework for the study of the cognitive bases of human technology only if we are able to verify some of its key facets. As suggested, we rewrote this part. We also rewrote the abstract and the first paragraph of the discussion.

“Here we focus on two key aspects of the technical-reasoning hypothesis that remain to be addressed: Generalizability and specificity. If technical reasoning is a specific form of reasoning oriented towards the physical world, then it should be implicated in *all* (the generalizability question) *and only* (the specificity question) the situations in which we need to think about the physical properties of our world.” (p. 5)

Introduction: The Goldenberg and Hagmann paper cited actually shows that familiar tool use may be based either on retrieval from semantic memory or by inferring function from structure (mechanical problem solving); in other words, the investigators saw a role for both kinds of information, and the relationship between mechanical problem solving and familiar tool use was actually relatively weak. This requires correction.

We disagree with Reviewer 3 on this point. The whole sentence is as follows:

“This silence has been initially broken by a series of studies initiated by Goldenberg and Hagmann (9), which has documented a behavioural link in left brain-damaged patients between common tool use and the ability to solve mechanical problems by using and even sometimes making novel tools (e.g., extracting a target out from a box by bending a wire to create a hook) (9, 17).” (p. 3-4)

We did not mention the interpretations given by Goldenberg and Hagmann about the link with the pantomime task, but only focused on the link they reported between common tool use and novel tool use. This is factual. In addition, we also disagree that the link between common tool use and novel tool use was weak.

“The hypothesis put forward in the introduction predicts that knowledge about prototypical tool use assessed by pantomime of tool use and the ability to infer function from structure assessed by novel tool selection can both contribute to the use of familiar tools. Indeed results of both tests correlated signicantly with the use of familiar tools pantomime of tool use: r = 0.77, novel tool selection: r = 0.62; both *P* < 0.001, but there was also a signicant correlation between the two tests r = 0.64, *P* < 0.001.” (Goldenberg & Hagmann, 1998; p. 585)

As can be seen in this quote, they reported a significant correlation between novel tool selection and the use of familiar tools. It is also noteworthy that the novel tool selection test and the pantomime test correlated together. Georg Goldenberg told one of the authors (F. Osiurak; personal communication) that this result incited him to revise its idea that pantomime could assess “semantic knowledge”, which explains why he did not use it again as a measure of semantic knowledge. Instead, he preferred to use a classical semantic matching task in his *2009 Brain* paper with Josef Spatt, in which they found a clearer dissociation between semantic knowledge and common/novel tool use not only at the behavioral level but also at the cerebral level.

Introduction: Please expand and clarify this sentence "However, this involvement seems to be task-dependent, contrary to the systematic involvement of left are PF. The IFG and LOTC activations observed in prior studies are of interest as well. Were they indeed all taskdependent in these studies?

We agree that this sentence is confusing. We meant that, in the studies reported just above in the paragraph, these regions were not systematically reported contrary to the left area PF. As we think that this information was not crucial for the logic of the paper, we preferred to remove it.

Introduction: If implicit mechanical knowledge is acquired through interactions with objects, how is that implicit knowledge conveyed to pass on the material culture to others?

We thank Reviewer 3 for this comment. Although mechanical knowledge is implicit, it can be indirectly transmitted to other individuals, as shown in two papers we published in Nature Human Behaviour (Osiurak et al., 2021) and Science Advances (Osiurak et al., 2022). Actually, verbal teaching is not the only way to transmit information. There are many other ways of transmitting information such as gestural teaching (e.g., pointing the important aspects of a task to make them salient to the learner), observation without teaching (i.e., when we observe someone unbeknown to them) or reverse engineering (i.e., scrutinizing an artifact made by someone else). We have shown that even in reverse-engineering conditions, participants can benefit from what previous participants have done to increase their understanding of a physical system. In other words, all these forms of transmission allow the learners to understand new physical relationships without waiting that these relationships randomly occur in the environment. There is a wide literature on social learning, which describes very well how knowledge can be transmitted without using explicit communication. In fact, it is very likely that such forms of transmission were already present in our ancestors, allowing them to start accumulating knowledge without using symbolic language. We did not add this information in the MS because we think that this was a little bit beyond the scope of the MS. Nevetheless, we cited relevant literature on the topic to help the reader find it if interested in the topic.

“Yet, recent accounts have proposed that non-social cognitive skills such as causal understanding or technical reasoning might have played a crucial role in cumulative technological culture (6, 29, 66). Support for these accounts comes from micro-society experiments, which have demonstrated that the improvement of technology over generations is accompanied by an increase in its understanding (67, 68), or that learners’ technical-reasoning skills are a good predictor of cumulative performance in such micro-societies (33, 69).” (p. 19)

What distinguishes this implicit mechanical knowledge from stored knowledge about object manipulation? Are these two conceptualizations really demonstrably (testably) different?

We agree that it is complex to distinguish between these two hypotheses as suggested by Seidel et al. (2023) cited above (see Reviewer 3 Point 8). We have conducted several studies to test the opposite predictions derived from each hypothesis. The main distinction concerns the understanding of physical materials and forces, which is central to the technical-reasoning hypothesis but not to the manipulation-based approach. Indeed, sensorimotor programs about tool manipulation are not assumed to contain information about physical materials and forces. In the present study, the understanding of physical materials and forces was needed in the four tasks hypothesized as requiring technical reasoning, i.e., the mechanical problem-solving task, the psychotechnical task and the PHYS-Only and INT+PHYS conditions of the mentalizing task. We can illustrate this aspect with items of each of these tasks. Figure 1A is of the mechanical problem-solving task.

As explained in the MS, participants had memorized the five possible tools before the scanner session. Thus, for 4 seconds, they had to imagine which of these tools could be used to extract the target out from the box. We did so to incit them to reason about mechanical solutions based on the physical properties of the problem. Then, they had 3 seconds to select the tool with the appropriate shape, here the right one. In this case, the motor action remains the same (i.e., pulling). Another illustration can be given, with the psychotechnical task (Figure 1B).

In this task, the participant had to reason as to whether the boat-tractor connection was better in the left picture or in the right picture. This needs to reason about physical forces, but there is no need to recruit sensorimotor programs about tool manipulation. Finally, a last example can be given with the PHYS-Only condition of the mentalizing task (but the logic is the same for the INT+PHYS condition except that the character’s intentions must also be taken into consideration Figure 1D).

Here the participant must reason about which picture shows what is physically possible. In this task, there is no need to recruit sensorimotor programs about tool manipulation. In sum, what is common between these three tasks is the requirement to reason about physical materials and forces. We do not ignore that motor actions could be simulated in the mechanical problemsolving task, but no motor action needed to be simulated in the other three tasks. Therefore, what was common between all these tasks was the potential involvement of technical reasoning but not of sensorimotor programs about tool manipulation. Of course, an alternative is to consider that motor actions are always needed in all the situations, including situations where no “manipulable tool” is presented, such as a tractor and a boat, a pulley, or a cannon. We cannot rule out this alternative, which is nevertheless, for us, prejudicial because it implies that it becomes difficult to test the manipulation-based approach as motor actions would be everywhere. We voluntarily decided not to introduce a debate between the reasoning-based approach and the manipulation-based approach and preferred a more positive writing by stressing the insights from the present study. Note that we stressed the merits of the manipulation-based approach in the introduction because we sincerely think that this approach has provided interesting insights. However, we voluntarily did not discuss the debate between the two approaches. Given Reviewer 3’s comment (see also Reviewer 1 Point 2), we understand and agree that some words must be nevertheless said to discuss how the manipulation-based approach could interpret our results, thus stressing the potential limitations of our interpretations. Therefore, we added a specific section in the discussion in which we discussed this aspect in more details.

“The second limitation concerns the alternative interpretation that the left area PF is not central to technical reasoning but to the storage of sensorimotor programs about the prototypical manipulation of common tools. Here we show that the left area PF is recruited even in situations in which participants do not have to process common manipulable tools. For instance, some items of the psychotechnical task consisted of pictures of tractor, boat, pulley, or cannon. The fact that we found a common activation of the left area PF in such tasks as well as in the mechanical problem-solving task, in which participants could nevertheless simulate the motor actions of manipulating novel tools, indicates that this brain area is not central to tool manipulation but to physical understanding. That being said, some may suggest that viewing a boat or a cannon is enough to incite the simulation of motor actions, so our tasks were not equipped to distinguish between the manipulation-based approach and the reasoning-based approach. We have already shown that the left area PF is more involved in tasks that focus on the mechanical dimension of the tool-use action (e.g., the mechanical interaction between a tool and an object) than its motor dimension (i.e., the interaction between the tool and the effector [e.g., 24, 40]). Nevertheless, we recognize that future research is still needed to test the predictions derived from these two approaches.” (p. 18-19)

Introduction and throughout: The framing of left Area PF as a special area for technical reasoning is overly reductionistic from a functional neuroanatomic perspective in that it ignores a large relevant literature showing that the region is involved with many other tasks that seem not to require anything like technical cognition. Indeed, entering the coordinates - 56, -29, 36 (reported as the peak coordinates in common across the studied tasks) in Neurosynth reveals that 59 imaging studies report activations within 3 mm of those coordinates; few are action-related (a brief review indicated studies of verbal creativity, texture processing, reading, somatosensory processing, stress reactions, attentional selection etc). Please acknowledge the difficulty of claiming that a large brain region should be labeled the brain's technical reasoning area when it seems to also participate in so much else. The left IPL (including area PF) is densely connected to the ventral premotor cortex, and this network is activated in language and calculation tasks as well as tool use tasks (e.g., Matsumoto, Nair, et al., 2012). What other constructs might be able to unite this disparate literature, and are any of these alternative constructs ruled out by the present data? Lacking this objective discussion, the manuscript does read as a promotion of the investigators' preferred viewpoint.

We thank Reviewer 3 for this comment. As stressed in the initial version of the MS, we did not write that the left area PF is sufficient but central to the network that allows us to reason about the physical world. Regardless, we agree that an objective discussion was needed on this aspect to help the reader not misunderstand our purpose. We added a section in this aspect as suggested.

“Before concluding, we would like to point out two potential limitations of the present study. The first limitation concerns the fact that the literature has documented the recruitment of the left area PF in many neuroimaging experiments in which there was no need to reason about physical events (e.g., language tasks). This can be easily illustrated by entering the left area PF coordinates in the Neurosynth database.

This finding could be enough to refute the idea that this brain area is specific to technical reasoning. Although this limitation deserves to be recognized, it is also true for many other findings. For instance, sensory or motor brain regions such as the precentral or the postcentral cortex have been found activated in many non-motor tasks, the visual word form area in non-language tasks, or the Heschl’s gyrus in nonmusical tasks. This remains a major challenge for scientists, the question being how to solve these inconsistencies that can result from statistical errors or stress that considerable effort is needed to understand the very functional nature of these brain areas. Thus, understanding that the left area PF is central to physical understanding can be viewed as a first essential step before discovering its fundamental function, as suggested by the functional polyhedral approach (56).” (p. 18)

Discussion: The discussion of a small cluster in the IFG (pars opercularis) that nearly survived statistical correction is noteworthy in light of the above point. This further underscores the importance of discussing networks and not just single brain regions (such as area PF) when examining complex processes. The investigators note, "a plausible hypothesis is that the left IFG integrates the multiple constraints posed by the physical situation to set the ground for a correct reasoning process, such as it could be involved in syntactic language processing". In fact, the hypothesis that the IFG and SMG are together related to resolving competition has been previously proposed, as has the more specific hypothesis that the SMG buffers actions and that the context-appropriate action is then selected by the IFG (e.g., Buxbaum & Randerath, 2018). The parallels with the way the SMG is engaged with competing lexical or phonological alternatives (e.g., Peramunage, Blumstein et al., 2011) have also been previously noted.

We added the Buxbaum and Randerath (2018)’s reference in this section.

“The functional role of the left IFG in the context of tool use has been previously discussed (24) and a plausible hypothesis is that the left IFG integrates the multiple constraints posed by the physical situation to set the ground for a correct reasoning process, such as it could be involved in syntactic language processing (for a somewhat similar view, see [51]).” (p. 16-17)

Introduction and Discussion: Please clarify how the technical reasoning network overlaps with or is distinct from the tool-use network reported by many previous investigators.

We added a couple of sentences in the discussion to clarify this point.

“It should be clear here that we do not advocate the localizationist position simply stating that activation in the left area PF is the necessary and sufficient condition for technical reasoning. We rather defend the view according to which it requires a network of interacting brain areas, one of them – and of major importance – being the left area PF. This allows the engagement of different configurations of cerebral areas in different technical-reasoning tasks, but with a central process acting as a stable component: The left area PF. Thus, when people intend to use physical tools, it can work in concert with brain regions specific to object manipulation and motor control, thereby forming another network, the tool-use network. It can also interact with brain regions specific to intentional gestures to form a “social-learning” network that allows people to enhance their understanding about the physical aspects of a technical task (e.g., the making of a tool) through communicative gestures such as pointing gestures (42). The major challenge for future research is to specify the nature of the cognitive process supported by the left area PF and that might be involved in the broad understanding of the physical world.” (p. 14)

Discussion: All of the experimental tasks require a response from a difficult choice in an array, and all of the tasks except for the fluid cognition task are likely to require prediction or simulation of a motion trajectory-whether an embodied or disembodied trajectory is unclear. The Discussion does mention the related (but distinct) idea of an "intuitive physics engine", a "kind of simulator", Please clarify how this study can rule out these alternative interpretations of the data. If the study cannot rule out these alternatives, the claims of the study (and the paper title which labels PF as a technical cognition area) should be scaled back considerably.

We thank Reviewer 3 for this comment. The authors of the papers on intuitive physics engine or associative learning do not suggest that these processes are embodied. As discussed above, we clarified our perspective on the role of the left area PF and hope that these modifications help the reader better understand it. We warmly thank Reviewer 3 for their comments, which considerably helped us improve the MS.